# Extensive and spatially variable within-cell-type heterogeneity across the basolateral amygdala

**Timothy P O'Leary[1†], Kaitlin E Sullivan[1†], Lihua Wang[2], Jody Clements[2], Andrew L Lemire[2], Mark S Cembrowski[1,2,3,4]***

[1]Department of Cellular and Physiological Sciences, Life Sciences Institute, University of British Columbia, Vancouver, Canada; [2]Janelia Research Campus, Howard Hughes Medical Institute, Ashburn, United States; [3]Djavad Mowafaghian Centre for Brain Health, University of British Columbia, Vancouver, Canada; [4]School of Biomedical Engineering, University of British Columbia, Vancouver, Canada

**Abstract** The basolateral amygdala complex (BLA), extensively connected with both local amygdalar nuclei as well as long-range circuits, is involved in a diverse array of functional roles. Understanding the mechanisms of such functional diversity will be greatly informed by understanding the cell-type-specific landscape of the BLA. Here, beginning with single-cell RNA sequencing, we identified both discrete and graded continuous gene-expression differences within the mouse BLA. Via in situ hybridization, we next mapped this discrete transcriptomic heterogeneity onto a sharp spatial border between the basal and lateral amygdala nuclei, and identified continuous spatial gene-expression gradients within each of these regions. These discrete and continuous spatial transformations of transcriptomic cell-type identity were recapitulated by local morphology as well as long-range connectivity. Thus, BLA excitatory neurons are a highly heterogenous collection of neurons that spatially covary in molecular, cellular, and circuit properties. This heterogeneity likely drives pronounced spatial variation in BLA computation and function.

**\*For correspondence:**
mark.cembrowski@ubc.ca

[†]These authors contributed equally to this work

**Competing interests:** The authors declare that no competing interests exist.

## Introduction

The amygdala is a brain region that governs a variety of functions and behaviors (*Janak and Tye, 2015*). Classically, the amygdala has been studied for the role it plays in acquisition and expression of conditioned fear memory (*Fanselow and LeDoux, 1999*; *Maren and Quirk, 2004*), with more recent evidence also implicating this brain region to be involved in other aversive states like anxiety (*Daviu et al., 2019*). Complementing this work examining the role of the amygdala in negative valence settings, it is also becoming increasingly apparent that the amygdala participates in appetitive and reward-based behavior (*Baxter and Murray, 2002*; *Wassum and Izquierdo, 2015*). The mechanisms by which this single brain region mediates such a variety of functions is unclear.

One reductionist approach to understanding amygdala computation lies in identifying functional contributions of specific amygdalar regions (*Beyeler and Dabrowska, 2020*). The basolateral amygdala complex (BLA) is one region that has received particular attention, in large part due to its high degree of reciprocal long-range connectivity with other brain regions (*Amir et al., 2018*; *Janak and Tye, 2015*; *Little and Carter, 2013*; *McGarry and Carter, 2017*; *Senn et al., 2014*). This anatomical arrangement suggests a powerful role for the BLA in orchestrating a variety of long-range computations, and moreover, enables circuit-specific experimental access for mapping circuits onto function and behavior (*Tovote et al., 2015*). Harnessing this experimental tractability, a body of work has emerged demonstrating that BLA projections to different downstream regions control of a range of

diverse, and sometimes bidirectional, functional and behavioral phenotypes (*Beyeler et al., 2018*; *Beyeler et al., 2016*; *Burgos-Robles et al., 2017*; *Felix-Ortiz et al., 2013*; *Felix-Ortiz et al., 2016*; *Felix-Ortiz and Tye, 2014*; *Herry et al., 2008*; *Kim et al., 2013*; *Namburi et al., 2015*; *Tye et al., 2011*).

One central consideration for interpreting circuit-based manipulations is understanding the extent to which BLA neurons can be considered to be an intrinsically homogeneous population (*Namburi et al., 2015*; *Zirlinger et al., 2001*). In principle, differential behavioral effects following circuit-specific BLA manipulations may simply reflect different downstream readouts of otherwise identical BLA inputs. Conversely, and not mutually exclusively, such differential behavior could also reflect variable intrinsic BLA neuron identity that conveys different long-range information independent of downstream readout. Indeed, as different BLA projections can emanate from different gross geographic locations (*Beyeler et al., 2018*; *McGarry and Carter, 2017*), there exists a potential organizational substrate wherein projection target and intrinsic identity of BLA neurons might spatially covary.

One perspective that can provide comprehensive insight into BLA neuronal identity is from transcriptomic 'cell typing', wherein cells with similar gene-expression properties can be identified and mapped onto higher order structural and functional roles (*Cembrowski, 2019*; *Lein et al., 2017b*; *Zeng and Sanes, 2017*). This approach has proven useful for deciphering cell-type-specific organization and operation in a wide variety of other brain regions, especially in cases where spatial variation in gene expression can be mapped to spatial variation in higher order properties (e.g. morphology, connectivity) (*Cembrowski et al., 2018a*; *Cembrowski and Spruston, 2019*; *Economo et al., 2018*; *Lein et al., 2017a*; *Mandelbaum et al., 2019*; *Phillips et al., 2019*; *Wu et al., 2017*). Thus, application of transcriptomic cell typing to the BLA has the potential to inform cell-type-specific identity rules and spatial organization, as well as to infer higher-order structural and functional correlates.

Here, we employed such a cell-typing approach to understand the extent and organization of heterogeneity within the excitatory neuronal population of the BLA. To initially assess variation within this population, we performed single-cell RNA sequencing, identifying both coarse discrete heterogeneity as well as fine-scale graded variation. To map this variation in space, we used single-gene and multiplexed in situ hybridization, mapping both discrete and graded transcriptomic variation onto spatial axes of the BLA. Finally, leveraging this spatial heterogeneity in gene expression, we identified similar spatial differences in BLA connectivity and morphology. Ultimately, our work demonstrates that excitatory neurons of the BLA are a highly heterogeneous collection of neurons that vary in space across molecular, cellular, and circuit properties. This prominent variation in intrinsic BLA identity likely contributes to pronounced circuit-specific and behavioural effects. To facilitate use of our data as a resource for future studies, we have created a web-based portal to allow easy access and analysis of the scRNA-seq data in our study (http://scrnaseq.janelia.org/amygdala; schematic: *Figure 1*).

## Results

### scRNA-seq reveals discrete gene-expression differences within the BLA

To perform scRNA-seq, we used our previously published manual approach, which facilitates capture of excitatory neurons due to their general abundance and post-dissociation viability (*Cembrowski et al., 2018a*; *Cembrowski et al., 2018b*). To capture BLA neurons, the BLA was microdissected, dissociated, and individual cells were manually obtained for scRNA-seq. After subsequent processing and sequencing (see Methods), high-depth scRNA-seq data was obtained for 1231 excitatory neurons (5.9 ± 1.2 thousand expressed genes/cell, mean ± SD). Broadly, these scRNA-seq profiles separated into two discretely separated groups via t-SNE visualization and principal component analysis, with this separation recapitulated by graph-based clustering (see Methods; *Figure 2A, B*). Such discrete separation was robust, as random classifiers trained on 100 randomly selected cells (~8.1% of total dataset) was sufficient to predict near-perfect classification accuracy of the remaining dataset (91.8% +/- 3.7% accuracy of remaining 1131 cells, N = 1000 trials, mean ± SD; *Figure 2C*). This discrete separation was also seen using UMAP dimensionality reduction (*Figure 2—figure supplement 1A,B*; *Becht et al., 2019*).

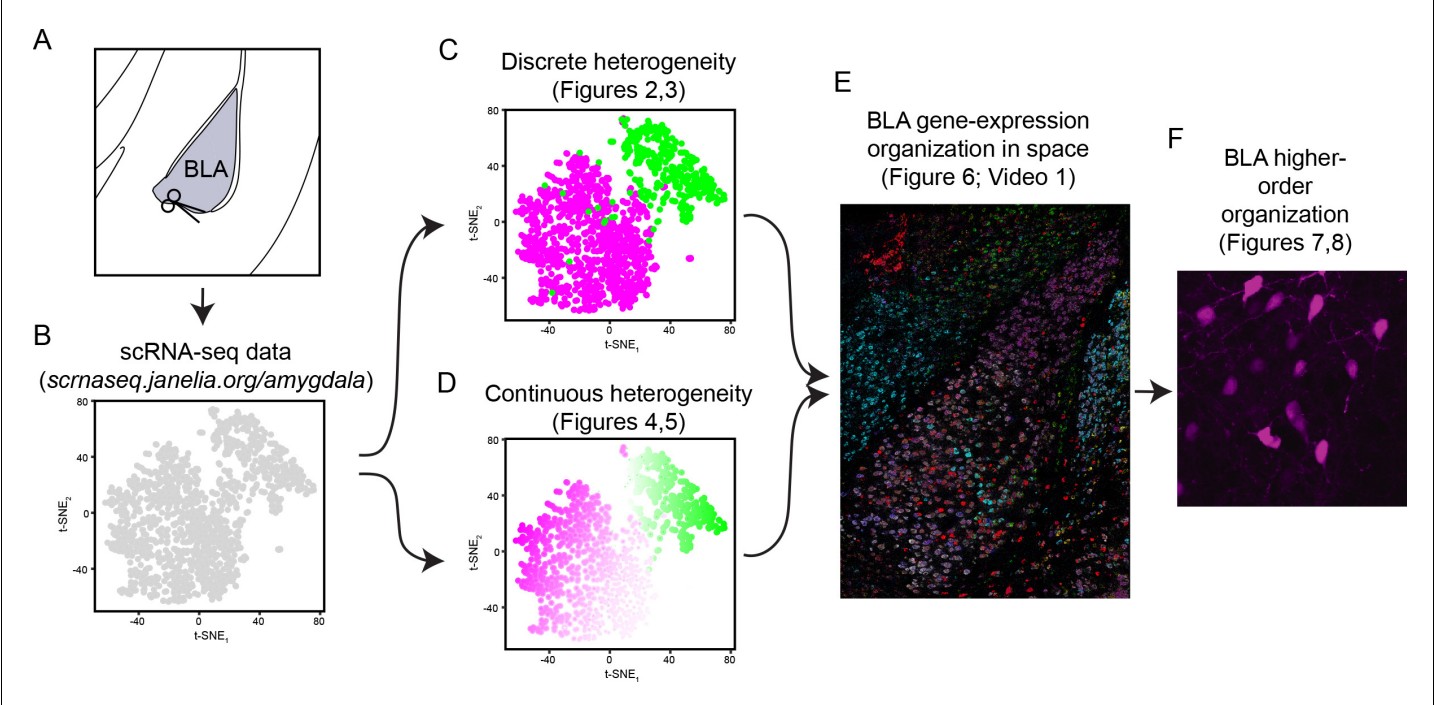

**Figure 1.** Workflow for assessing heterogeneity within the basolateral amygdala complex. (**A**) Atlas schematic of the basolateral amygdala complex (BLA), schematizing microdissection for scRNA-seq. (**B**) Overview of scRNA-seq data, as visualized through t-SNE dimensionality reduction. Data, along with analysis and visualization tools, available at http://scrnaseq.janelia.org/amygdala. (**C**) Analysis of coarse discrete heterogeneity within the BLA (see *Figures 2* and *3*). (**D**) Analysis of fine continuous heterogeneity within the BLA (see *Figures 4* and *5*). (**E**) Spatial registration of discrete and continuous heterogeneity (see *Figure 6*). (**F**) Higher order discrete and continuous heterogeneity within the BLA (see *Figures 7* and *8*).

Broadly, sequenced cells expressed markers of neurons in general (e.g. *Snap25)*, as well as more specific markers of excitatory neurons (e.g. *Slc17a7*, encoding Vglut1, and *Camk2a*; *Figure 2D*). As expected, expression of markers for inhibitory neurons (e.g., *Gad1* and *Slc32a1*) was effectively absent. Thus, both coarse clusters of BLA cells represented excitatory neuronal populations. Such excitatory subtypes of BLA cells were not resolved in a recently published relatively low-depth drop-let-based scRNA-seq dataset (*Zeisel et al., 2018*) (see *Figure 2—figure supplement 2*), highlighting the utility of our high-depth manual capture approach.

We next examined gene expression that differentiated the two clusters of excitatory neurons. Using an adjusted *p*-value threshold of 0.05, we identified a total of 415 differentially expressed genes between the two clusters. This differential expression incorporated genes that were relatively binary in expression ('on' vs. 'off': e.g., *Cplx1* and *Negr1*; *Figure 2E*), as well as genes that exhibited expression in only a subset of neurons in the enriched cluster (e.g. *Col5a2*, *Calb1*; *Figure 2E*). More-over, these differentially expressed genes encompassed a wide array of functionally relevant categories for neurons, including axon guidance and cell adhesion, ligands and receptors, calcium handling, synapses, and transcriptional regulation (*Figure 2F*). These results illustrate many differentially expressed genes across two BLA excitatory neuron subtypes, and suggest that this differential expression likely maps on to higher-order functional variability.

## Discrete separation in gene expression maps onto the lateral vs. basal amygdala nuclei

We next mapped the spatial location of the two discrete clusters. To do this, we analyzed cluster-specific marker gene expression via chromogenic in situ hybridization (ISH) data from the Allen Mouse Brain Atlas (*Lein et al., 2007*; *Table 1*). Marker gene expression was examined across the anterior-posterior axis of the BLA in coronal sections (*Figure 3A–D*), with the spatial extent of the BLA identifiable by *Slc17a7* expression (*Figure 3E–F*).

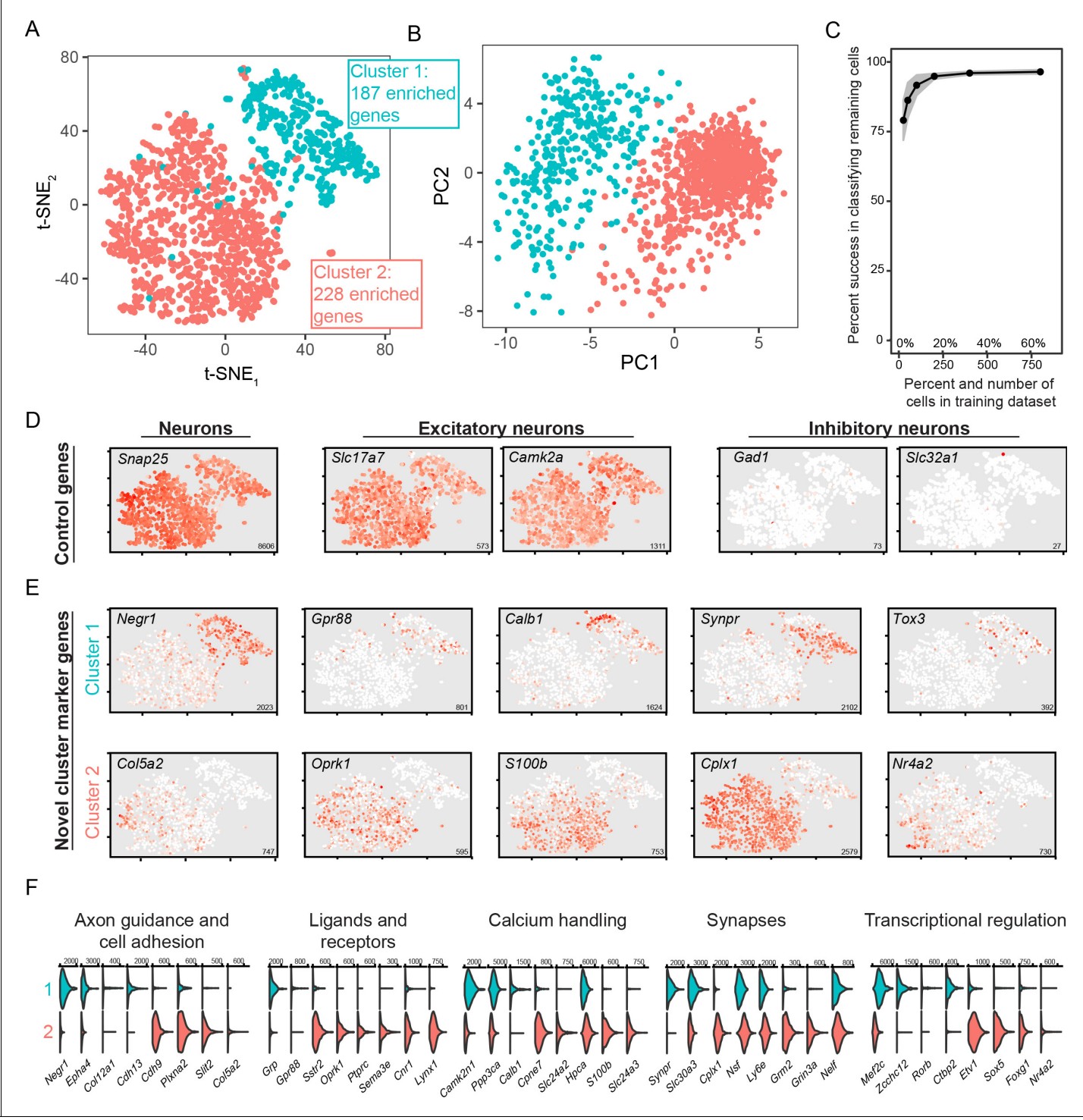

**Figure 2.** scRNA-seq analysis of coarse, discrete transcriptomic differences. (**A**) Overview of scRNA-seq data, as visualized through t-SNE dimensionality reduction and colored according to cluster identity. The number of enriched genes for each cluster is provided. (**B**) As in (**A**), but with projections onto the first two principal components. (**C**) Random forest classification of subsampled data. (**D**) Expression of known control genes. Expression is colored from low (white) to high (red). Inset numerical values denote maximum CPM value across all cells. (**E**) As in (**D**), but with novel cluster-enriched marker genes. (**F**) Functionally relevant neuronal genes that are differentially expressed between clusters. Numerical values denote CPM values of right tick mark.

The online version of this article includes the following figure supplement(s) for figure 2:

**Figure supplement 1.** UMAP and t-SNE dimensionality reduction recapitulate the same general organization.

*Figure 2 continued on next page*

*Figure 2 continued*

**Figure supplement 2.** Comparison with previously published BLA scRNA-seq data.

Marker genes for cluster 1 (e.g. *Negr1*, *Ddit4l*) were generally enriched in posterior sections, occupying the lateral-dorsal regions of the BLA (*Figure 3G–H*). Notably, such spatial enrichment corresponds to the lateral amygdala (LA; *Figure 3D*). Conversely, expression of marker genes for cluster 2 (e.g. *Cplx1*, *Lynx1*) was generally present in the basal amygdala (BA), abutting and largely nonoverlapping with expression for cluster one markers (*Figure 3I–J*). Thus, spatial registration of cluster 1 and cluster 2 indicated that these clusters corresponded to the LA and BA, respectively. Consistent with this, the relative representation of the two clusters in the scRNA-seq dataset was similar to the relative abundance of LA and BA neurons counted from ISH (scRNA-seq: 27% and 73% from LA and BA neurons respectively, cf. 39% and 61% from ISH using atlas delineation of LA vs. BA).

We note that some marker genes can indeed show expression in the opposite nuclei, albeit at a markedly reduced density (e.g. BA-enriched *Cplx1* is sparsely expressed in the LA, *Figure 3I*). Such lower expression density in the depleted region was similar to the overall abundance of interneurons (*Figure 3—figure supplement 1A,B*), suggesting that such geographical expression 'spill-over' reflects interneuron labeling rather than displaced excitatory neurons. In support of this, many cluster markers showed minimal spill-over expression (e.g. *Ddit4l*, *Lynx1*: *Figure 3E,G*; additional marker genes: *Figure 3—figure supplement 1C–F*), with two-color ISH illustrating this spill-over predominantly reflected *Gad1*-expressing interneurons (see *Figure 3—figure supplement 2*).

## Graded heterogeneity within the basal and lateral amygdala

Next, we examined the existence and organization of further fine-scale heterogeneity within each cluster. To begin, we allowed finer clustering of our scRNA-seq dataset, which split the LA into two subpopulations and the BA into four subpopulations (*Figure 4A*; see Materials and methods). Such subpopulations were generally abutting and/or interspersed when projections onto the first two principal components were examined (*Figure 4B*), as well as with UMAP dimensionality reduction (*Figure 2—figure supplement 1C,D*), suggesting that these subpopulations might reflect graded differences within the LA and BA rather than discrete subtypes. Reinforcing this, random forest classification was relatively error-prone, with neurons that occupied the interface of within-LA and within-BA subpopulations being poorly classified in particular (*Figure 4C*).

We therefore considered the possibility that these subpopulations reflected graded, rather than discrete, differences (*Cembrowski and Menon, 2018*). We first pursued this by examining heterogeneity within the LA, leveraging the fact that gene expression between the two subpopulations varied substantially (e.g. 199 genes differentially expressed at $p_{adjusted}$ <0.05; *Figure 4D*). Enriched genes within a given subpopulation exhibited a high degree of gene-to-gene variation, wherein the extent of expression of one marker gene was a poor predictor of the extent of expression in another marker gene (*Figure 4E,F*). Moreover, when comparing between the two subpopulations, differentially expressed genes did not exhibit a clear reciprocal boundary that classically embodies discrete separation (*Cembrowski and Menon, 2018*; *Figure 4E*). All of these observed signatures are suggestive of LA heterogeneity existing in a continuous spectrum, rather than adhering to discretely separated populations.

In a similar fashion, we considered the four BA subpopulations, examining the 147 genes that were enriched within individual subpopulations (*Figure 4G*). As with the LA, these enriched genes exhibited high gene-to-gene variability within a subpopulation, and as well lacked reciprocal boundaries between subpopulations (*Figure 4H,I*). These findings, coupled with the fact that these subpopulations were abutting and/or interleaved in dimensionally reduced spaces (*Figure 4A,B*) and exhibited poor separation at interfaces (*Figure 4C*), suggested that borders between subpopulations of the BA were graded rather than discrete.

## Spatial variation within the lateral and basal amygdala

Motivated by scRNA-seq-graded variation within LA and BA clusters, we next sought to examine whether such gradients exhibited a spatial correlate. We began with the LA, examining ISH for markers of the two LA subpopulations (*Figure 5A*). Markers for the two subpopulations were

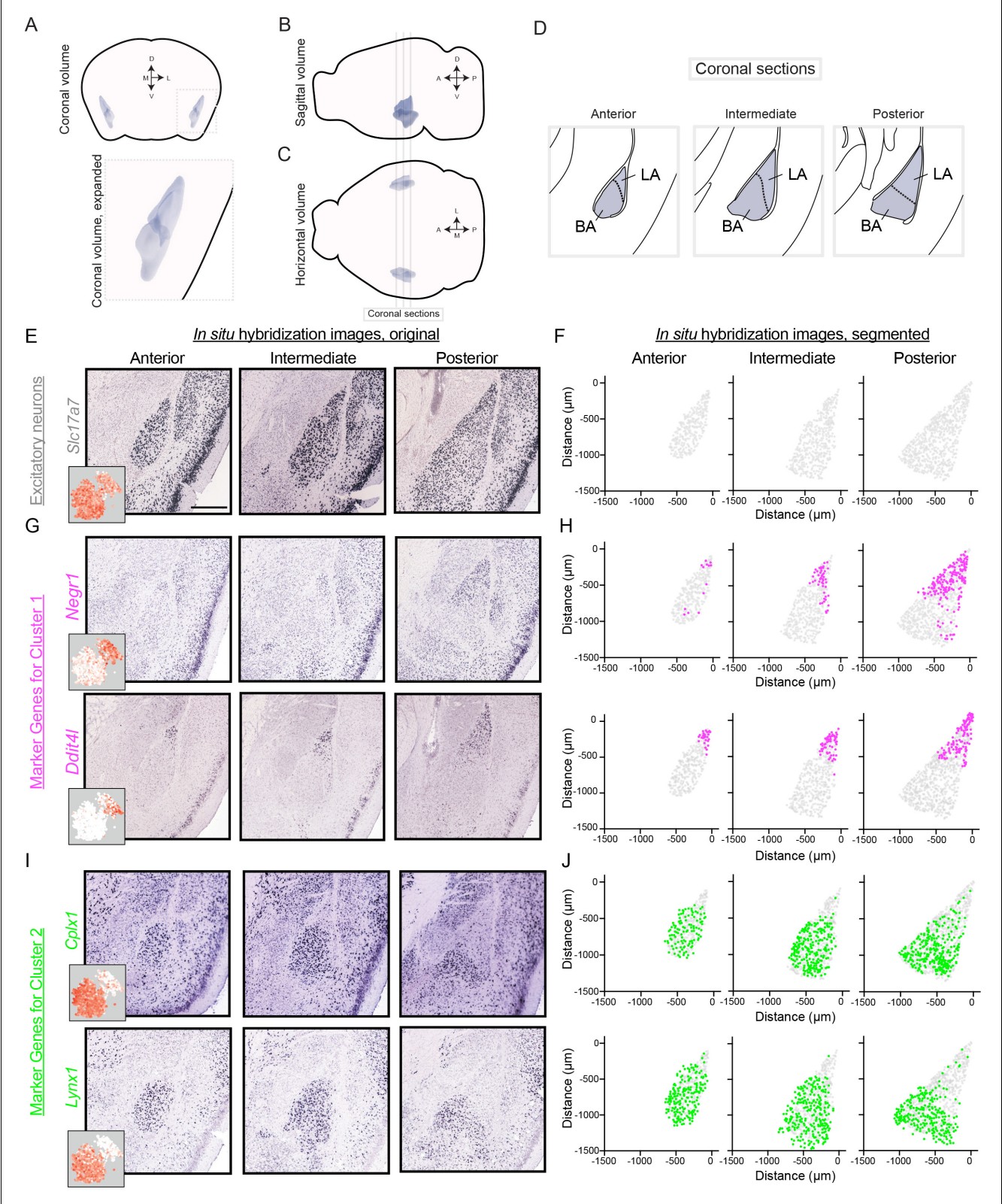

**Figure 3.** Differentially expressed genes show spatially discrete variation within the BLA. (A) Three-dimensional coronal volumetric rendering of the BLA in the mouse brain. Image modified from Allen Brain Explorer (*Lau et al., 2008*). (B,C) As in (A), but for sagittal (B) and horizontal (C) views of the mouse brain. Coronal sections defined in (D) are shown. (D) Coronal sections highlighting the anterior, intermediate, and posterior geometry of the BLA. Atlas definitions of LA vs. BA are shown (modified from *Paxinos and Franklin, 2004*). (E) Spatial expression of *Slc17a7* via chromogenic ISH across

*Figure 3 continued on next page*

Figure 3 continued

the anterior, intermediate, and posterior BLA. Scale bar: 500 μm. Inset shows scRNA-seq expression. (F) Segmentation of *Slc17a7*-expressing cells from the BLA. (G,H) As in (E,F), but for *Negr1* and *Ddit4l,* novel markers of cluster 1. For reference, locations of segmented cells are superimposed on the location of excitatory (*Slc17a7*-expressing) neurons within the BLA. (I,J) As in (G,H), but for *Cplx1* and *Lynx1,* novel markers of cluster 2.

The online version of this article includes the following figure supplement(s) for figure 3:

**Figure supplement 1.** Spatial organization of inhibitory marker gene expression, and additional cluster-specific marker gene expression.

**Figure supplement 2.** Cplx1 expression in the lateral amygdala is associated with inhibitory neurons.

enriched in grossly different regions of the LA: one set of subpopulation markers (e.g. *Rorb, Myl4*) was relatively enriched in the anterior and dorsal LA, whereas markers for the other subpopulation (e.g. *Cdh13, Otof*) were relatively enriched in posterior and ventral LA (**Figure 5B**). As expected from scRNA-seq, strict spatial boundaries between subpopulation marker genes were not apparent; similarly, prominent gene-to-gene variability in spatial expression within a given subpopulation was also present. In sum, such results suggest graded spatial variation of cell-type identity within the LA.

Similar spatial organizational rules, consistent with graded cell-type identity, were present within the BA (**Figure 5C,D**). Marker genes for BA subpopulation one were generally enriched anterior and medial, whereas markers for BA subpopulation two were enriched in more posterior and lateral regions of the BA. Markers for BA subpopulation 3 and 4 showed spatial variation, wherein subpopulation three was enriched medially and subpopulation four was enriched laterally. As with the LA, no sharp spatial delineations were seen between subpopulations, and marker genes associated with the same BA subpopulation spatially varied on a gene-to-gene basis.

Importantly, two features of this variable cell identity suggest that gene expression differences can be largely attributed to graded baseline differences in cell type, rather than cell-state differences driven by activity. First, most enriched genes (**Figure 4D–I**) are not associated with activity dependence. Second, for the genes that have been associated with activity dependence (e.g. *Bdnf, Nr4a2*), they exhibit different patterns of expression across cells (*Bdnf* vs. *Nr4a2:* scRNA-seq: **Figure 4H**, ISH: **Figure 5C,D**; see also **Figure 6—figure supplement 1** for within-sample mFISH comparison). Although we cannot completely discount activity states driving some gene-expression differences within the BLA population, the above results argue that activity-driven cell-state differences are small relative to baseline cell-type variability.

## Multiplexed spatial mapping captures graded cell-type identities

From single-gene ISH, both BA and LA scRNA-seq heterogeneity can be mapped onto spatial gradients of cell-type identity. However, mapping single genes within individual sections means that gene-expression covariation within individual cells is lost. Additionally, chromogenic ISH is qualitative in nature, which precludes quantitative analyses of gene-expression. To circumvent these limitations, and simultaneously investigate both discrete and continuous heterogeneity with the BLA, we next employed multiplexed single-molecule fluorescent in situ hybridization (mFISH). With this multiplexed approach, we mapped 12 gene-expression targets in the same tissue, providing a comprehensive assessment of cell-type identity and organization at single-cell resolution in space. For gene selection, we used *Slc17a7* to identity excitatory neurons, *Negr1* and *Cplx1* to study the putative discrete spatial divide between LA and BA, and the subpopulation marker genes *Nr4a2, Otof, Cdh13, Rorb, Adamts2, Prss23, Bdnf, Slc5a5,* and *Nnat* (**Figures 2–5**). Quantitative data on the expression of each gene was obtained on a per-cell basis, via calculating the number of signal-containing pixels normalized by the cell area (see Materials and methods).

Gene expression was mapped in anterior, intermediate, and posterior BLA sections (**Figure 6A–C**, **Figure 6—figure supplement 1**, **Video 1**). To initially study putative discrete differences between LA and BA, we derived a 'phenotype index' that corresponded to the relative expression of LA marker gene *Negr1* and BA marker gene *Cplx1* (defined on a per-cell basis according to $(E_{Cplx1}-E_{Negr1})/(E_{Cplx1}+E_{Negr1})$, where $E$ is the expression for the indicated gene). With this metric, cells exclusively expressing *Negr1* have phenotypic indices of −1, cells exclusively expressing *Cplx1* have indices of 1, and cells expressing non-zero amounts of both genes occupy intermediate values. Using this metric to study BLA phenotypes across space, it was apparent that phenotypes sharply

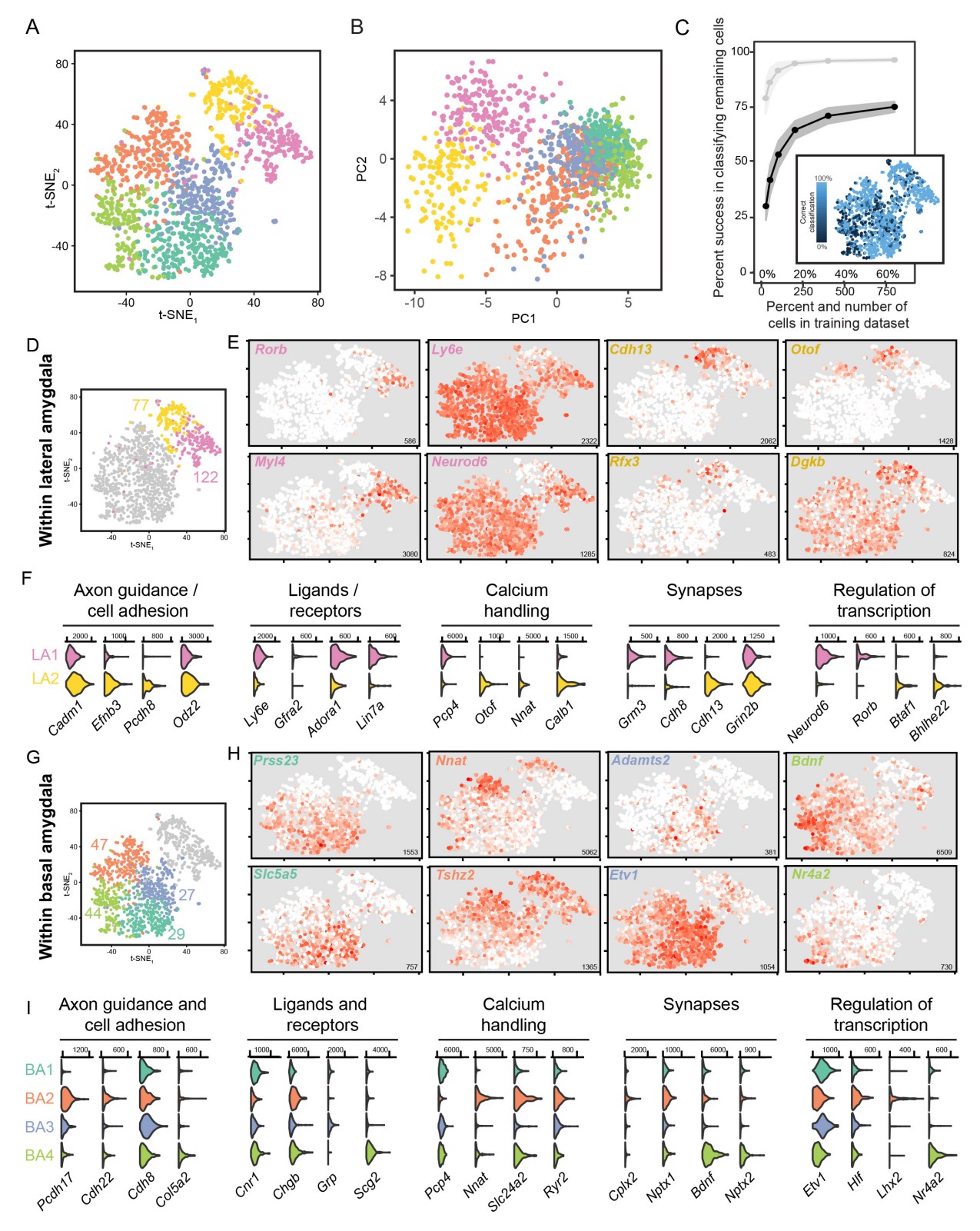

**Figure 4.** scRNA-seq analysis of fine, graded transcriptomic differences. (**A**) Overview of scRNA-seq data, as visualized through t-SNE dimensionality reduction, at a relatively fine clustering resolution. (**B**) As in (**A**), but with projections onto the first two principal components. (**C**) Random forest classification of subsampled data. For comparison, results from the coarse 2-cluster scheme are also provided in light grey. Inset illustrates rates of misclassification on a per-cell basis, with misclassified cells typically found at the interface of fine clusters. Inset results depict averages across 1000

*Figure 4 continued on next page*

*Figure 4 continued*

simulations for 800 training cells. (**D**) t-SNE illustration of fine-scale clustering within the LA, with number of enriched genes shown for each subpopulation. (**E**) Expression of example marker genes for the two subpopulations of LA neurons. Note high gene-to-gene variation in expression for marker genes associated with each subpopulation. Inset numerical values denote maximum CPM value across all cells. (**F**) Expression of functionally relevant genes that are enriched in each LA subpopulation. (**G,H,I**) As in (**D,E**), but for subpopulations of the BA.

transitioned across the LA-BA border (*Figure 6D*), with single-cell phenotypes exhibiting a bimodal distribution concentrated at extreme phenotypic index values (*Figure 6E*).

We next used mFISH to visualize graded spatial changes in gene-expression and cell-type identity within the LA and BA. To do this, a given seed cell was selected (e.g. magenta cell, *Figure 6F*), and correlations of mFISH-detected gene expression were computed to all remaining cells (*Figure 6F*; see Materials and methods). Distances between the seed cell and all remaining cells within the same section were calculated, with all cells assigned a LA or BA discrete phenotype (according to a winner-take-all binarization of the results of *Figure 6D,E*). This approach allowed the spatial profile of gene-expression correlation to be computed for a given seed cell, and to be examined relative to phenotypic identity (*Figure 6G*; see also *Figure 6H,I*). Averaging this process across all cells, we identified a spatial gradient of gene expression, providing direct evidence for graded cell-type identity within BA and LA (*Figure 6J*). Finally, clustering of mFISH data at different resolutions recapitulated both the discrete and graded cell-type identity changes, providing further evidence of these organizational patterns within the BLA (*Figure 6—figure supplement 2*).

## scRNA-seq and ISH map interneuron subtypes and geography

In principle, the scRNA-seq and ISH approaches used previously may also provide insight into the inhibitory cell-type organization of the BLA. To examine this, we next analyzed the 51 interneuron single-cell transcriptomes obtained by scRNA-seq (*Figure 6—figure supplement 3A–C*). Despite the small number of cells in this dataset, this analysis identified two distinct subtypes of interneurons that differentially expressed dozens of genes. Via chromogenic ISH, we found these subtypes occupied distinct regions, with one subtype within the BLA and the other occupying the pericapsular region (*Figure 6—figure supplement 3D,E*). Such results illustrate that different subtypes of neurons occupy the pericapsular and interior regions of the BLA, and highlight that our high-read-depth approach can identify heterogeneity despite surveying relatively few cells.

## Long-range projections recapitulate organization of spatial heterogeneity

As a focal point of BLA research involves long-range circuits of excitatory neurons (*Janak and Tye, 2015*), we next examined whether spatial gene-expression patterns we observed in the BLA mapped to specific circuits. To do so, we used the retrograde AAV tracer rAAV2-retro (*Tervo et al., 2016*), and in a given animal injected fluorescent tracers (either rAAV2-retro-GFP or rAAV2-retro-tdT) into downstream targets of the BLA. In total, six projections were examined: auditory cortex (ACX), nucleus accumbens (NAC), medial entorhinal cortex (MEC), ventral hippocampus (VHC), prefrontal cortex (PFC), and the retrosplenial cortex (RSC).

Projections to the ACX and NAC recapitulated the discrete LA-BA divide that was identified transcriptomically, such that ACX projections were enriched within the LA and NAC projections where enriched in the BA (*Figure 7A–H*; both obeyed $p < 0.001$ for enrichment relative to random selected BLA neurons in $N = 1000$ Monte-Carlo simulations). Similarly, the remaining four projections – MEC, VHC, PFC and RSC – all were enriched with the BA (*Figure 7I–X*; $p < 0.001$ via $N = 1000$ Monte-Carlo simulations). These projections furthermore exhibited spatially restricted regions reminiscent of BA subpopulation marker gene expression (*Figure 7—figure supplement 1*; $p < 0.001$ via $N = 1000$ Monte-Carlo simulations when comparing to cell-to-cell distances expected by random selection of BA neurons). Thus, spatial heterogeneity was present for each BLA projection examined, consistent with organizational rules observed at a transcriptomic level.

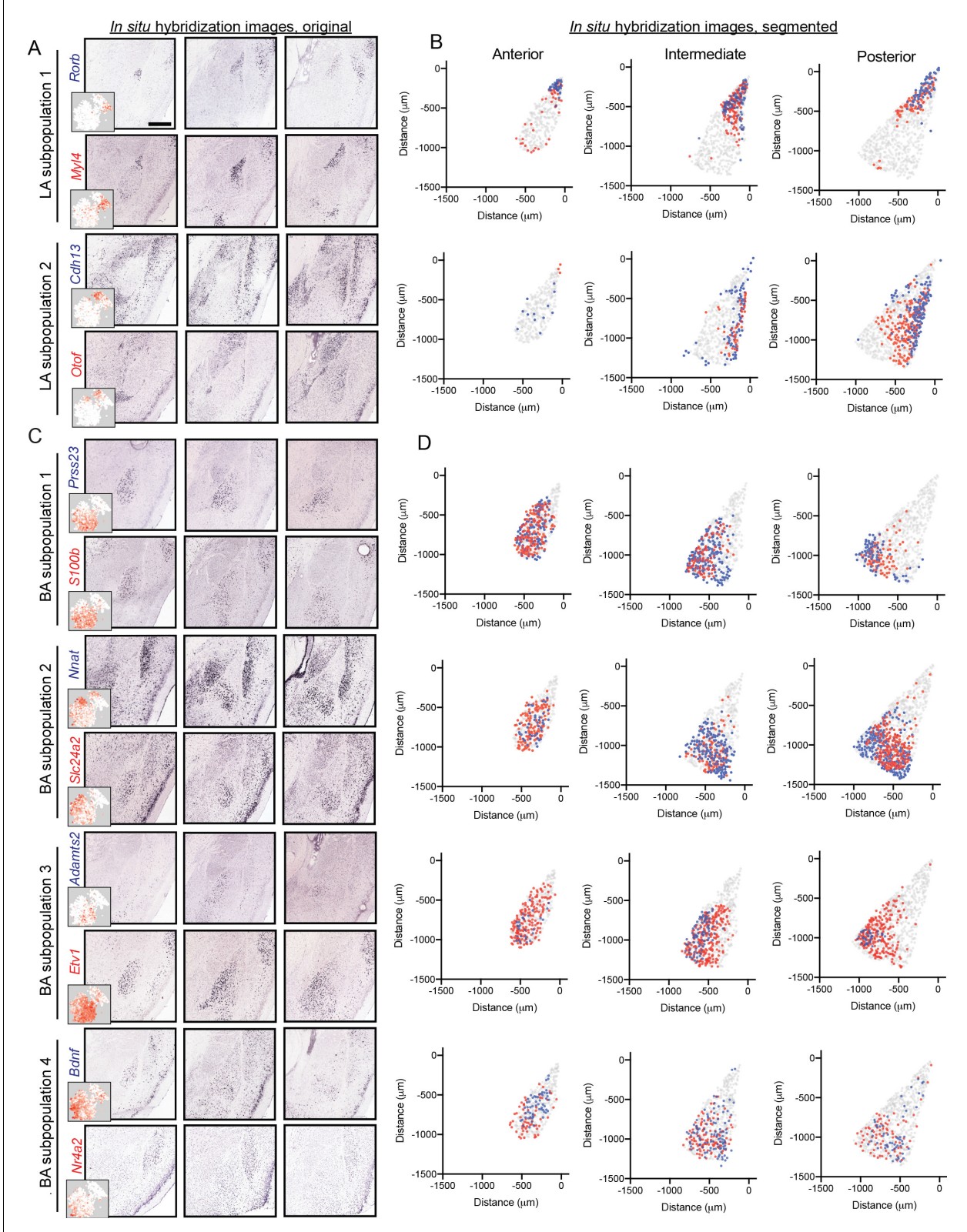

**Figure 5.** Graded spatial variation, within the LA and BA, of subpopulation-enriched marker genes. (**A**) Representative ISH images for selected LA subpopulation-enriched genes. Scale bar: 500 μm. Insets show t-SNE visualization of scRNA-seq data for each gene. (**B**) Locations of segmented cells for pairs of genes for each LA subpopulation across anterior, intermediate, and posterior sections. For reference, locations of cells are superimposed on the location of excitatory (*Slc17a7*-expressing) neurons within the BLA. (**C,D**) As in (**A,B**), except for pairs of enriched genes for BA subpopulations.

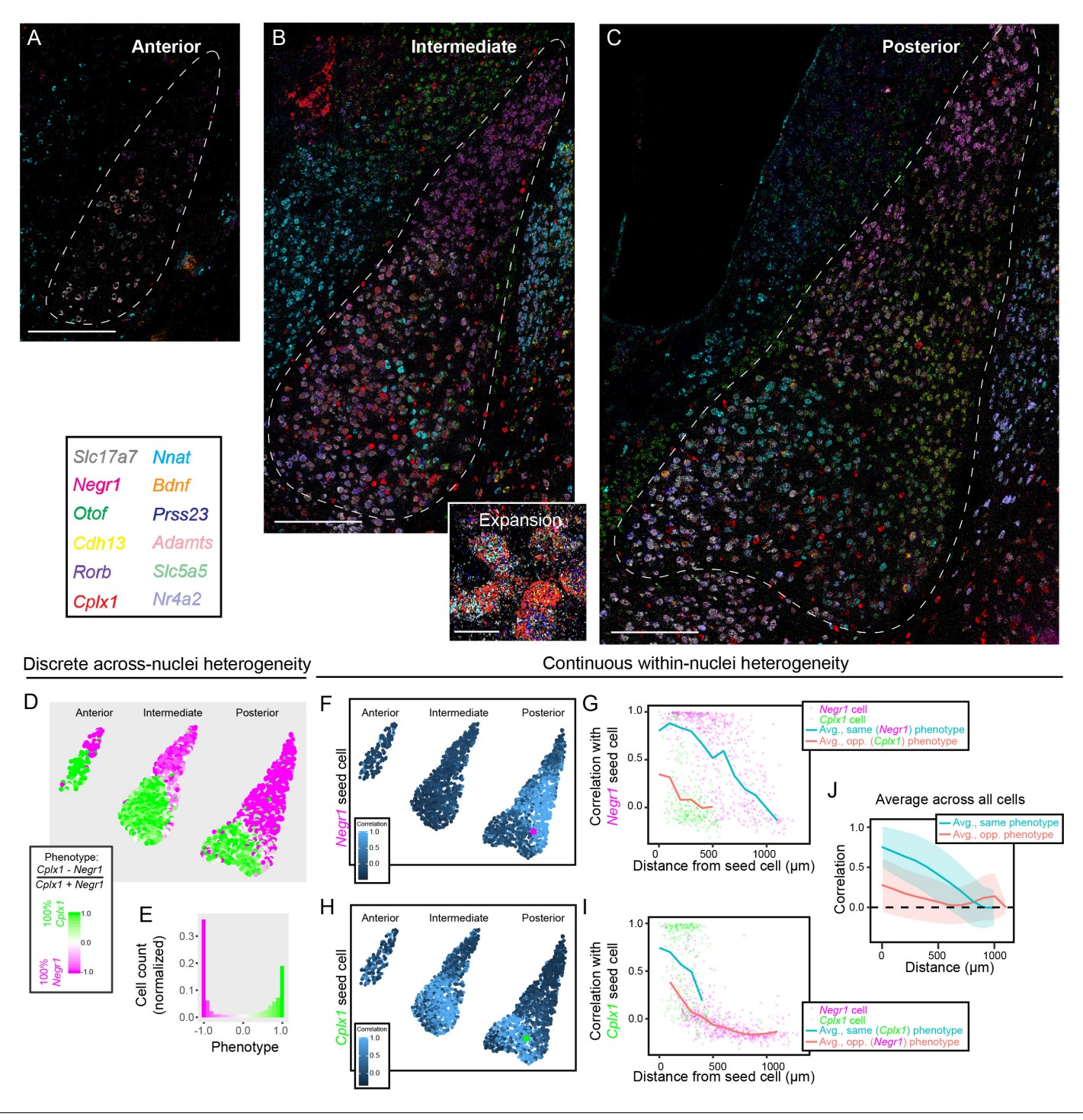

**Figure 6.** Multiplexed FISH simultaneously maps discrete and continuous heterogeneity. (**A**) Overview of mFISH in the anterior amygdala. Scale bar: 400 µm. (**B**) As in (**A**), but for intermediate amygdala. Inset provides representative expansion, scale bar: 20 µm. (**C**) As in (**B**), but for posterior amygdala. (**D**) Overview of discrete heterogeneity in the BLA, obtained by assessing relative expression of *Negr1* and *Cplx1* across cells. (**E**) Histogram of phenotype identities for (**D**). (**F**) Illustration of gene-expression correlations for all cells, relative to the *Negr*1 phenotype seed cell highlighted in magenta. (**G**) Correlations computed and averaged across space, relative to the magenta seed cell shown in (**F**). Magenta and green points represent other cells within the posterior BLA with a binarized discrete cell-type identity. Blue and orange lines indicate binned average correlation across all cells at 100 µm spacing for the same and opposite phenotypes. (**H,I**) As in (**F,G**), but for a *Cplx1* phenotype seed cell. (**J**) As in (**G,I**), but pooling and analyzing averages across all cells in the BLA. Blue and orange lines illustrate mean correlations to other cells of the same and opposite phenotypes, respectively, with spread about mean indicating one standard deviation.

*Figure 6 continued on next page*

*Figure 6 continued*

The online version of this article includes the following figure supplement(s) for figure 6:

**Figure supplement 1.** Individual mFISH marker-gene expression in the BLA.
**Figure supplement 2.** Clustering of mFISH excitatory neurons.
**Figure supplement 3.** Interneuron subtypes resolved by scRNA-seq and chromogenic ISH.

## Discrete morphological differences across the LA-BA axis

Finally, we leveraged our previous assays to see whether differences within cellular properties could be observed across the LA-BA axis. In particular, we considered whether there is a difference in cell-body size across the LA-BA border. Identifying such differences is important, as cell-body size can strongly shape passive electronic properties, the influence of active channels in the membrane, and biophysical processes that are shaped by surface area-to-volume ratios (*Dayan and Abbott, 2001*).

To begin, we examined cell-body area in our projection level dataset (N = 3–5 mice per injection site, 12–82 cells/mouse). In doing so, we identified a precipitous decline in the cell body size of ACX-projecting neurons, relative to projections from the VHC, MEC, PFC, NAC and RSC (p<0.005 via Mann-Whitney U test; *Figure 8A,B*). Of note, this drop covaried with the LA vs. BA projection site of these neurons. For pairwise comparisons within the BA (VHC, MEC, PFC, NAC, RSC), we saw no statistical differences in cell-body size (p>0.05 via Mann-Whitney U test for all pairwise within-BA comparisons).

The previous results are shaped by the particular projections we examined, and thus, we next sought to take a complementary unbiased approach to survey cell-body area of LA vs. BA neurons. From our mFISH experiments (*Figure 6*), we extracted channels that corresponded to markers of excitatory neurons (*Slc17a7*), LA neurons (*Negr1*), and BA neurons (*Cplx1*) (*Figure 8C*). Leveraging the fact that dense *Slc17a7* expression effectively acted as a cell-body fill, we examined cell-body areas of *Slc17a7*-expressing neurons that were either *Negr1*-expressing or *Cplx1*-expressing (see Materials and methods) (*Figure 8D*). In doing so, we again recapitulated a marked difference in cell body size between LA and BA excitatory neurons (p<$2.2 \times 10^{-16}$ via Mann-Whitney U Test, *Figure 8E*), further underscoring that a discrete cell-body size difference exists between LA and BA excitatory neurons.

## Discussion

### LA and BA as distinct entities

Frequently, the LA and BA are grouped together as elements of the basolateral amygdala complex (BLA). This aggregation is typically due to a lack of a clear cytoarchitectonic boundary between the two nuclei in most experimental settings (e.g. DAPI counterstain). As this LA-BA divide can be challenging to practically identify, this may also suggest that the LA and BA can be functionally aggregated. However, given that the amygdala is frequently referenced as a brain region that is relatively arbitrarily named in both structure and function (*Swanson and Petrovich, 1998*), such a BLA gross aggregation may conceal important underlying heterogeneity.

Indeed, a collection of disparate evidence illustrates specific features that divide the LA and BA. Classically, the LA and BA have been partitioned with respect to histochemical stains (e.g. acetylcholinesterase) and the orientation of fibers (*LeDoux, 2007*). There is a variety of work that shows LA and BA also exhibit differences in gene expression (*Zirlinger et al., 2001*), morphology (*McDonald, 1984*; *Sah et al., 2003*), and behavioral recruitment (*Beyeler et al., 2018*; *Herry et al., 2008*). Although these results argue for differences between the LA and BA, their disparate and multimodal nature make it challenging to assess the overall degree of separability between the LA and BA. It is also challenging to assess whether such heterogeneity emerges from continuous or discrete differences across the two nuclei.

An advantage of our transcriptomics-based analyses is that it allows complete quantification of a given feature; that is, quantified expression of every gene in the genome. Leveraging this complete approach, our work revealed hundreds of differentially expressed genes between the LA and BA, producing discrete separation between the two nuclei in both gene-expression space and

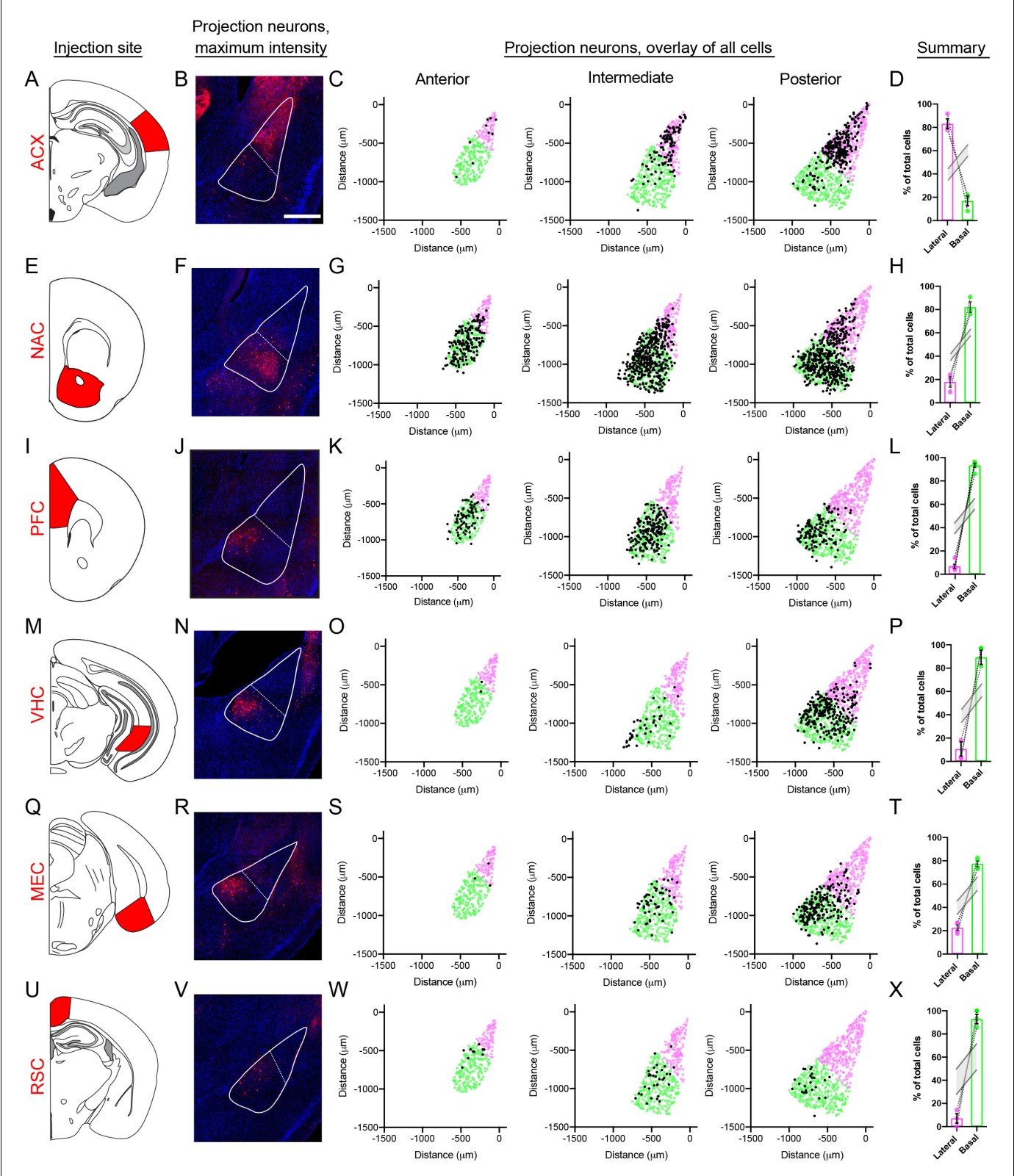

**Figure 7.** BLA projection neurons show spatially discrete and continuous cellular variation. (**A**) Atlas schematic showing retrograde viral injection site for the ACX (modified from *Paxinos and Franklin, 2004*). (**B**) Representative maximum intensity projections of labeled cells within the BLA. Scale bar: 500 µm. The boundary of the BLA is shown (solid line) along with the division between the lateral and basal sub-regions (dashed line) (**C**). Locations of all virally labeled cells along the A/P axis of the BLA (left to right). For reference, virally labeled cells (black) are superimposed over the position of

*Figure 7 continued on next page*

*Figure 7 continued*

excitatory neurons (*Slc17a7*-expressing cells), with cells colored to illustrate LA (magenta) and BA (green) per atlas definitions. (D) Percentage of total labeled cells located within the LA and BA, depicted as mean ± SEM. Grey lines indicate range of values within 95% confidence intervals for the expected number of cells in the LA and BA, via Monte Carlo simulations of random selection of excitatory neurons. E-X. As in (A–D), except for the NAC (E–H), PFC (I–L), VHC (M–P), MEC (Q–T) and RSC (U–X) injection sites.

The online version of this article includes the following figure supplement(s) for figure 7:

**Figure supplement 1.** Spatial cellular gradients of projection neurons are present in the BLA and differ based on projection site.

geographical space (*Figures 2*, *3* and *6*). In addition to this degree of differential expression being numerically large, many of these genes were also central to neuronal structure and function, thus suggesting that higher order structural and functional differences should be apparent within the BLA. This prediction was embodied by both differences in projection targets (*Figure 7*) and cell-body size (*Figure 8*). Taken in concert with previous findings, this work argues that excitatory neurons of the BA vs. LA should be considered as intrinsically distinct entities (summary: *Figure 9*).

## Graded spatial heterogeneity and functional implications

Our work here revealed pronounced graded transcriptomic variability in LA and BA excitatory neurons (*Figure 4*). In principle, discrete separation of BA subtypes may emerge with sampling additional cells; however, high-depth scRNA-seq data requires relatively few cells to resolve discrete subtypes (*Figure 2C*; *Figure 2—figure supplement 2*; *Figure 6—figure supplement 3*; *Cembrowski et al., 2018b*; *Erwin et al., 2020*). Thus, this graded transcriptomic identity seems to be a *bona fide* feature within the LA and BA.

Consistent with this, complementary techniques showed spatial gradients of marker-gene expression (*Figure 6*). Such a spatial arrangement is not necessarily expected a priori; indeed, there is evidence in many other brain regions wherein prominent graded transcriptomic heterogeneity does not have a spatial correlate (*Harris et al., 2018*; *Stanley et al., 2020*). The finding of a spatial covariate of this gene-expression heterogeneity is important, as it facilitates histological cross-validation of scRNA-seq data (*Figures 5* and *6*). Moreover, it provides a 'Rosetta stone' for registering and comparing our work to previous spatially resolved structural and function characterizations (*Cembrowski and Spruston, 2019*).

In atlas parcellations of the BLA, both the BA and LA are frequently partitioned into subnuclei, with the LA divided into the dorsolateral, ventrolateral, and ventromedial domains, and the BA divided into anterior and posterior domains. Although such discrete parcellations are provided to emphasize differences between these geographic regions, a body of evidence suggests that the heterogeneity underlying these subnuclei divisions may be spatially graded rather than discrete. For example, dendritic morphologies change gradually across subnuclei (*McDonald, 1984*; *Sah et al., 2003*), both the BA and LA have continua in firing patterns (*Sah et al., 2003*), and topographical gradients of connectivity are present across the BLA (*Beyeler et al., 2018*). Similar functional correlates, although typically sampled at a relatively coarser resolution in space, are also present within the BA along the anterior-posterior axis (*Bergstrom et al., 2013*; *Goosens and Maren, 2001*; *Kim et al., 2016*).

What are the potential functional advantages of such graded heterogeneity? One such answer may be found when considering this organization with respect to the computational demands required of the BLA. In particular, this brain region subserves neuronal and behavioural responses to fear, anxiety, and reward (*Baxter and Murray, 2002*; *Janak and Tye, 2015*; *Maren and Quirk, 2004*). In all these settings, graded neural representations are likely important for ensuring graded behavioral responses, and having a continuum in BA and LA cell-type identity could facilitate population-level coding of graded neural representations and responses. Indeed, such an organizational scheme is consistent with postulated graded representations within the BLA populations (*Kyriazi et al., 2018*), and thus such graded identity may allow cell-type variability to help match the space of necessary computations.

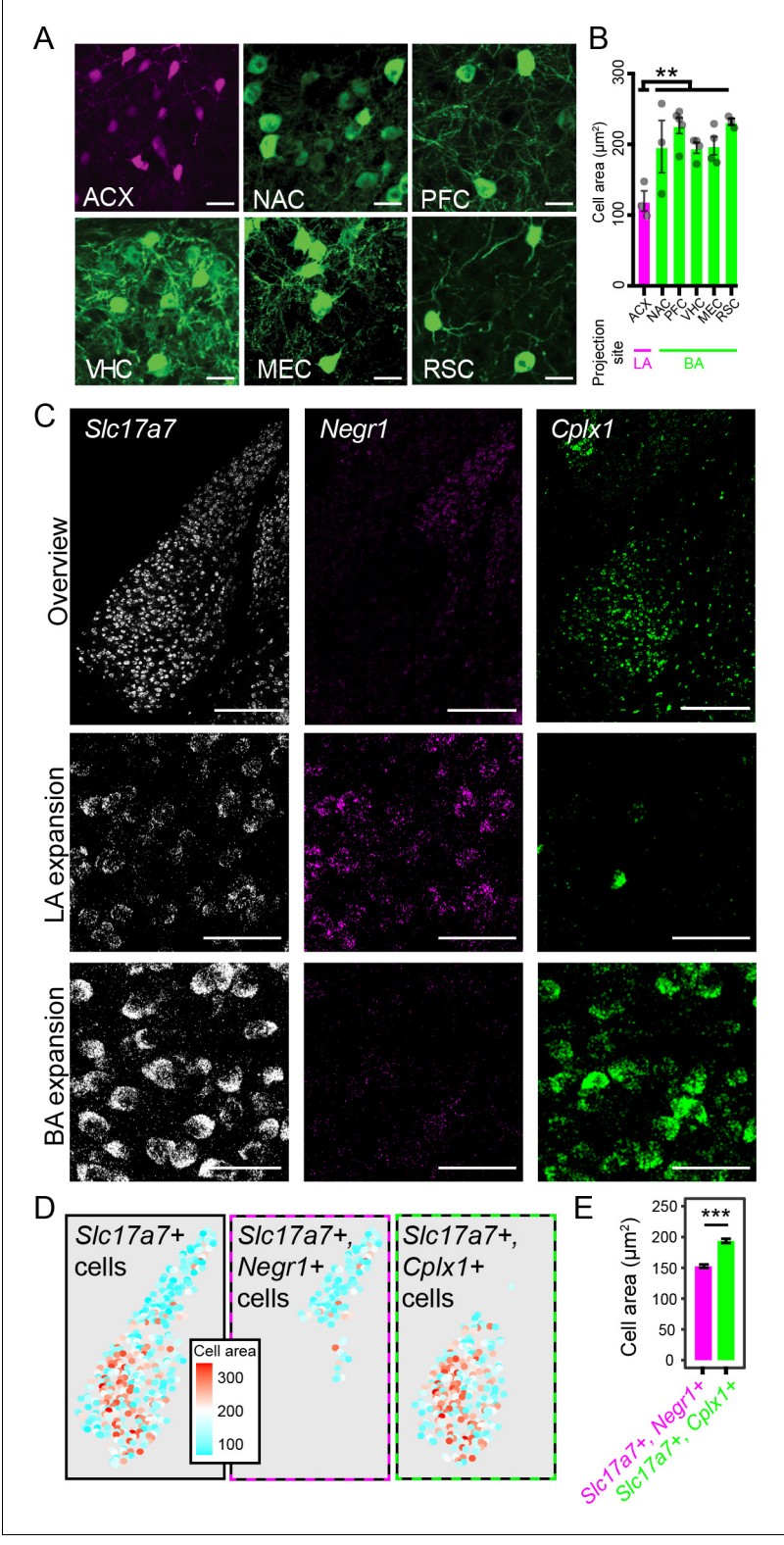

**Figure 8.** LA and BA neurons vary by morphology. (**A**) Representative images of labeled cells projecting to the ACX, NAC, PFC, VHC, MEC, and RSC. Scale bars: 25 μm. (**B**) The cell-body area of virally labeled cells for each projection site, depicted as mean ± SEM. (**C**) Top row: overview of mFISH signals for *Slc17a7*, *Negr1*, and *Cplx1* in the intermediate BLA. Scale bars: 300 μm. Middle, bottom rows: expansions for the LA and BA, respectively. Scale bars: 40 μm. (**D**) Left: Cell-body area of *Slc17a7*-expressing cells segmented from (**C**). Middle, right: Cell-body area

*Figure 8 continued on next page*

*Figure 8 continued*

of *Slc17a7*-expressing cells that also express *Negr1* or *Cplx1*, respectively. (**E**) Cell-body area for *Slc17a7*-expressing cells that also express *Negr1* (magenta) or *Cplx1* (green), depicted as mean ± SEM.

## Cell-type classification: transcriptomics vs other methodologies

As our findings here leverage transcriptomics for cell-type classification, it is important to make the distinction between transcriptomically defined cell types relative to cell types defined by other criteria (*Zeng and Sanes, 2017*). Of particular note to the graded transcriptomic identity found here, cells that have a continuous gene-expression identity may nonetheless be functionally distinct in ways that are not predicted solely by transcriptomic organization (*Kim et al., 2020*). One such example of this is a gene-expression threshold needed for associated protein products and function, which can effectively impose a higher-order discrete phenotype (*Cembrowski and Menon, 2018*). Thus, although transcriptomic cell-type identities constrain and inform higher order classification, a perfect correspondence between transcriptomic and higher-order identities is not a necessity.

**Table 1.** List of image series used from the Allen Mouse Brain Atlas.

| Gene | Figure | Image series |
| --- | --- | --- |
| Kcng1 | 2-S2C | 77340480 |
| Negr1 | 2-S2F, 3D | 692 |
| Cplx1 | 2-S2I, 3F, 6-S1D | 67752308 |
| Strip2 | 2-S2L | 72283809 |
| Zbtb20 | 2-S2O | 79568020 |
| Slc17a7 | 3B | 70436317 |
| Ddit4l | 3D | 71836878 |
| Lynx1 | 3F | 655 |
| Gad1 | 3-S1B | 79556706 |
| Slc32a1 | 3-S1B | 72081554 |
| Cpne8 | 3-S1E | 73520974 |
| Gpr88 | 3-S1E | 79567811 |
| Rspo2 | 3-S1H | 71016632 |
| Cdh9 | 3-S1H | 72472764 |
| Rorb | 5B | 79556597 |
| Myl4 | 5B | 72129251 |
| Cdh13 | 5B | 79490066 |
| Otof | 5B | 73788043 |
| Prss23 | 5E | 70634118 |
| S100b | 5E | 79591593 |
| Nnat | 5E | 77887874 |
| Slc24A2 | 5E | 71924238 |
| Adamts2 | 5E | 71924385 |
| Etv1 | 5E | 72119595 |
| Bdnf | 5E | 79587720 |
| Nr4a2 | 5E | 732 |
| Slc6a1 | 6-S1D | 79591685 |
| Tshz1 | 6-S1D | 72129289 |
| Meis2 | 6-S1D | 1231 |

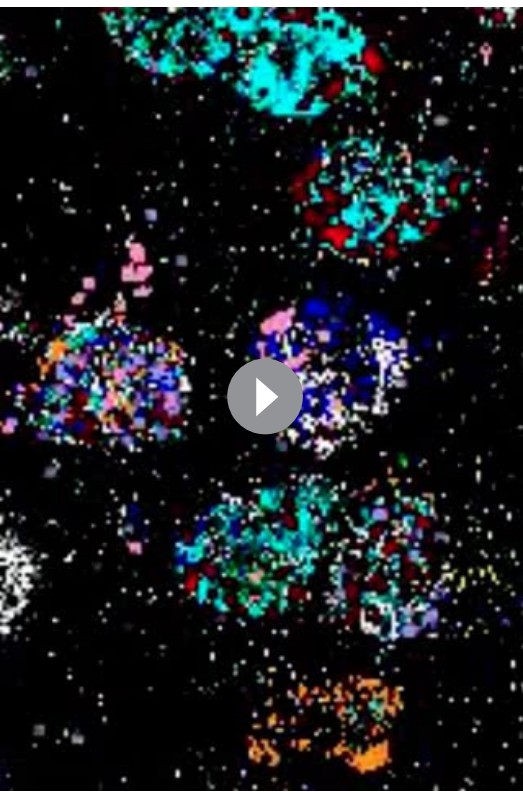

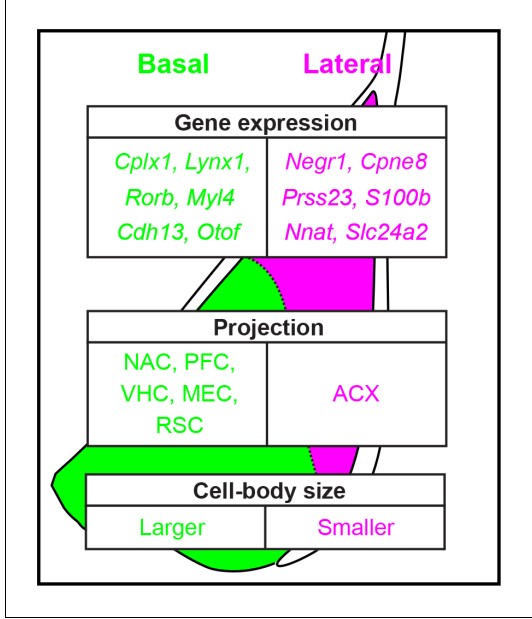

**Figure 9.** Summary of heterogeneity within the basolateral amygdala. Examples of spatially variable gene-expression, circuit, and cellular properties of excitatory neurons in the basolateral amygdala, as resolved in this study.

**Video 1.** Overview of mFISH marker gene expression in the posterior BLA.
https://elifesciences.org/articles/59003#video1

This point has particular relevance for consideration of within-BA differences, wherein published work argues for the discrete separation within the BA nucleus. Here, magnocellular and parvocellular populations have been proposed (*Alheid, 2003*). Putative markers for these populations, *Ppp1r1b* and *Rspo2*, have been identified from bulk microarray analysis and associated with different behavioral recruitment and function (*Kim et al., 2016*). Within our scRNA-seq data, *Ppp1r1b* and *Rspo2* are enriched in opposite ends of the BA transcriptomic spectrum and do not conform to discrete subtypes. Thus, *Ppp1r1b and Rspo2* populations may reflect graded subtypes at a transcriptomic level, with biological nonlinearities producing apparently step-wise differences in high-order function and behavior.

### Implications for circuit manipulations

In previous work, projections from the LA and BA have been associated with different long-range targets (*Beyeler et al., 2018*; *Hoover and Vertes, 2007*; *McGarry and Carter, 2017*; *Reppucci and Petrovich, 2016*; *Senn et al., 2014*; *Tsukano et al., 2019*; *Yang et al., 2016*). At a finer spatial scale within the BA, gradients that vary upon projection site have also been reported (*Beyeler et al., 2018*; *McGarry and Carter, 2017*). Our study, revealing discrete and continuous differences in cell-type identity (*Figures 2–6*) that differentially map onto these long-range targets (*Figure 7*), shows that neurons comprising these projections encompass an intrinsically diverse collection of cells.

As long-range projections are frequently leveraged for BLA circuit-specific manipulations, our work here emphasizes that care should be taken to avoid confounds when comparing manipulations of different circuits. In particular, prominent heterogeneity within the BLA excitatory neuron population indicates that manipulations of different circuits target BLA neurons that can vary in intrinsic properties. In principle, such heterogeneity may make manipulation efficacy inherently variable between projections, especially for methodologies that rely on intrinsic properties of neurons to exert effects (e.g. DREADDs: *Roth, 2016*). A means of circumventing such limitations, and complementing existing circuit-specific manipulations, lies in examining intrinsic activity of the BLA across space agnostic to projection target. Such an approach leverages the spatial heterogeneity identified

in our study, and will help to identify how spatial gene-expression differences map onto functional variability. In combination within circuit-specific manipulations, this work will provide an important understanding of how intrinsic- and circuit-level operations interact to evoke the wide array of BLA-associated phenotypes (*Janak and Tye, 2015*).

### Informing and interpreting future experiments

Our work here has revealed the spatial transcriptomic organization of excitatory neurons in the BLA. For future studies, this spatial framework can effectively act as a 'Rosetta Stone', wherein different spatially resolved experimental modalities can be compared and integrated. In addition to this correlative work, our work here also reveals genes that can be leveraged to access and manipulate BLA cells (e.g. via transgenic mice), as well as molecular targets that can be manipulated to causally relate specific gene-expression products to computation and function (e.g. via CRISPR-Cas). Thus, our findings here will help to inform a broad range of observational and interventional experiments in future work.

To facilitate the continued use of our data, we have hosted our scRNA-seq data online in conjunction with analysis and visualization tools (http://scrnaseq.janelia.org/amygdala). This web portal will help guide further exploration and hypothesis generation from these complex data, with associated tools allowing this analysis to be conducted in a straightforward and intuitive fashion. The combination of this accessibility and utility will help further exploration of cellular and molecular heterogeneity of the BLA, and how this heterogeneity relates to computation, function, and behavior.

## Materials and methods

**Key resources table**

| Reagent type (species) or resource | Designation | Source or reference | Identifiers | Additional information |
|---|---|---|---|---|
| Sequence-based reagent | *Cplx1* ISH probe | Advanced Cell Diagnostics | 482531-C3 | Two-color FISH |
| Sequence-based reagent | *Gad1* ISH probe | Advanced Cell Diagnostics | 400951-C2 | Two-color FISH |
| Sequence-based reagent | *Nr4a2* ISH probe | Advanced Cell Diagnostics | 423351-T1 | Multiplexed FISH |
| Sequence-based reagent | *Otof* ISH probe | Advanced Cell Diagnostics | 485671-T2 | Multiplexed FISH |
| Sequence-based reagent | *Cdh13* ISH probe | Advanced Cell Diagnostics | 443251-T3 | Multiplexed FISH |
| Sequence-based reagent | *Rorb* ISH probe | Advanced Cell Diagnostics | 444271-T4 | Multiplexed FISH |
| Sequence-based reagent | *Adamts2* ISH probe | Advanced Cell Diagnostics | 806371-T5 | Multiplexed FISH |
| Sequence-based reagent | *Prss23* ISH probe | Advanced Cell Diagnostics | 447921-T6 | Multiplexed FISH |
| Sequence-based reagent | *Bdnf* ISH probe | Advanced Cell Diagnostics | 424821-T7 | Multiplexed FISH |
| Sequence-based reagent | *Slc5a5* ISH probe | Advanced Cell Diagnostics | 487721-T8 | Multiplexed FISH |
| Sequence-based reagent | *Slc17a7* ISH probe | Advanced Cell Diagnostics | 416631-T9 | Multiplexed FISH |
| Sequence-based reagent | *Nnat* ISH probe | Advanced Cell Diagnostics | 432631-T10 | Multiplexed FISH |
| Sequence-based reagent | *Negr1* ISH probe | Advanced Cell Diagnostics | 806361-T11 | Multiplexed FISH |
| Sequence-based reagent | *Cplx1* ISH probe | Advanced Cell Diagnostics | 482531-T12 | Multiplexed FISH |

*Continued on next page*

*Continued*

| Reagent type (species) or resource | Designation | Source or reference | Identifiers | Additional information |
|---|---|---|---|---|
| Software, algorithm | R | https://www.r-project.org | SCR_001905 | - |
| Software, algorithm | Seurat | https://satijalab.org/seurat/ | RRID:SCR_007322 | - |
| Software, algorithm | Fiji | https://imagej.net/Fiji | RRID:SCR_002285 | - |
| Software, algorithm | Prism | https://www.graphpad.com/scientific-software/prism/ | RRID:SCR_002798 | - |
| Other | AAV-SL1-CAG-tdT | Janelia Viral Core | - | - |
| Other | AAV-SL1-CAG-GFP | Janelia Viral Core | - | - |
| Strain, strain background *Mus musculus* | Penk-cre | Janelia Research Campus | - | - |

Experimental procedures were approved by the Animal Care Committee at the University of British Columbia and the Institutional Animal Care and Use Committee at the Janelia Research Campus.

### Single-cell RNA sequencing data acquisition and analysis

The single-cell RNA-seq dataset (5.9 ± 1.2 thousand expressed genes/cell from 123 ± 65 thousand reads/cell, mean ± SD) was generated according to a previously published protocol (*Cembrowski et al., 2018a*; *Cembrowski et al., 2018b*). To capture individual neurons from the basolateral amygdala (BLA), brain sections were obtained from five mature ($\geq$8 weeks of age) male C57BL/6 mice. Three of these mice were WT, whereas the remaining two two mice were double-transgenic Penk2-cre x Ai14 mice (*Madisen et al., 2010*), used to leverage tdTomato expression to visually identify the BLA to facilitate complete microdissection (*Oh et al., 2014*). Cells extracted from both genetic backgrounds clustered together, indicating no effect of genetic background nor incomplete microdissection of BLA in WT mice. In all cases, the BLA was microdissected and dissociated, with manual purification (*Hempel et al., 2007*) used to capture cells and place into eight-well strips. For all datasets, library preparation, sequencing, and initial count-based quantification (*Dobin et al., 2013*; *Trapnell et al., 2009*) was performed according to previous methods (*Cembrowski et al., 2018a*). No blinding or randomization was used for the construction or analysis of this dataset. No a priori sample size was determined for the number of animals or cells to use; note that previous methods have indicated that several hundred cells from a single animal is sufficient to resolve heterogeneity within excitatory neuronal cell types (*Cembrowski et al., 2018a*; *Cembrowski et al., 2018b*). Raw and processed scRNA-seq datasets have been deposited in the National Center for Biotechnology Information (NCBI) Gene Expression Omnibus under GEO: GSE148866.

Computational analysis was performed in R (RRID:SCR_001905) (*R Development Core Team, 2008*) using a combination of Seurat (RRID:SCR_007322) (*Satija et al., 2015*) and custom scripts (*Cembrowski et al., 2018a*). Cells with <10,000 total counts were excluded from analysis (n = 69 of 1370 initial cells). For all remaining cells, counts were converted to Counts Per Million (CPM) for subsequent analysis. Putative non-neuronal cells (n = 19) were eliminated from the dataset by rejecting cells that exhibited CPM < 250 for *Snap25*, a pan-neuronal marker. For examining excitatory neurons (*Figures 2–5*), interneurons (n = 51) were eliminated from the dataset by rejecting cells that exhibited CPM > 100 for *Gad1*, an interneuron marker. Variable genes (n = 108) used for PCA were obtained with Seurat via *FindVariableGenes(mean.function = ExpMean, dispersion.function = LogVMR, x.low.cutoff = 0.125, x.high.cutoff = 3, y.cutoff = 1)*. Clusters were obtained with Seurat via *FindClusters(reduction.type = 'pca', dims. use = 1:10)*, using *resolution = 0.2* to obtain two coarse discrete clusters and *resolution = 0.8* to examine finer-scale heterogeneity. For

examining interneurons (*Figure 6—figure supplement 3*), 51 interneurons were retained (via CPM > 100 for *Gad1*), and variable genes and clusters were obtained via *x.low.cutoff = 0.0125*, *x. high.cutoff = 3*, *y.cutoff = 3*, *resolution = 1*, and *dims. use = 1:2* parameters. In all cases, subpopulation-specific enriched genes obeying $p_{ADJ} < 0.05$ were obtained with Seurat via *FindMarkers()*, where is the $p_{ADJ}$ is the adjusted p-value from Seurat based on Bonferroni correction. Functionally relevant differentially expressed genes were obtained using *FindMarkers()*, allowing for both cluster-specific enriched and depleted genes obeying $p_{ADJ} < 0.05$. t-SNE visualization *van der Maaten and Hinton, 2008* used perplexity = 30 (interneuron analysis: 20), with 1000 iterations (sufficient for convergence) on the default seed. Qualitatively similar results were obtained for other seed values. UMAP visualization (*Becht et al., 2019*) was obtained via the R UMAP package, using default parameters, performed on the scaled Seurat dataset with highly variable genes. In total, this approach and associated parameters provided discrete clusters that were consistent with dimensionally reduced visualizations and robust to downsampling, and predicted organizations that were validated by complementary histological methodologies (*Cembrowski and Spruston, 2017*).

When plotting gene expression using t-SNE, color ranges from white (zero expression) to red (maximal expression), plotted logarithmically, with the maximum CPM value across all cells provided as an inset. For random forest classification (*ClassifyCells()* in Seurat), random subsets of graph-based clustered cells were taken (n = 50, 100, 200, 400, or 800 cells; n = 100 random subsets for each number of cells), and used to predict the cluster identities of the remaining cells in the dataset.

To compare our work to a recently published scRNA-seq dataset (*Zeisel et al., 2018*), we downloaded the 'TEGLU22 (Excitatory neurons, amygdala)' dataset from this publication (14,897 total cells). A minimum of 500 total counts was required to initially retain 13,826 total cells from this dataset (1.2 ± 0.7 thousand genes expressed/cell). Interneurons and non-neuronal cells were excluded via the same thresholds used in our scRNA-seq data, and excitatory neurons were further selected for by requiring CPM > 100 for the excitatory neuronal marker *Slc17a7* for these remaining cells, resulting in 1975 total cells used for analysis. Subsequent analysis and visualization, including extraction of variable genes and clusters, was performed identically to our in-house dataset.

## Chromogenic in situ hybridization

All chromogenic ISH images were obtained from the publicly available Allen Mouse Brain Atlas (AMBA) (*Lein et al., 2007*). At least one coronal section was selected for analysis within each of the anterior (−0.82 to −0.94 from bregma), intermediate (−1.34 to −1.70 from bregma) and posterior (−1.82 to −2.06 from bregma) regions of the BLA. To facilitate segmentation of expressing cells, images were processed with a gaussian blur filter, binarized with a manually determined intensity threshold, and adjacent cells separated with a watershed function. The locations of segmented cells between 100–400 μm$^2$ were then measured within the BLA.

To spatially register cells to a common orientation, the BLA within each image was traced using six points, and then rotated and translated onto a BLA template using a procrustes transformation in R without scaling (*Beyeler et al., 2018*). The transformation was then applied to the location of segmented cells for the respective image, and then segmented cells were plotted relative to the dorsal-most point of the template. In a minority of cases, a small translation of segmented cells (typically <200 μm) was subsequently performed manually if the procrustes translation misaligned BLA borders. The number of cells within different sub-regions of the BLA were obtained after transformation of segmented cells. Sub-regional boundaries were created for each of the anterior, intermediate or posterior BLA templates, and analyses of all images within a respective BLA anterior-to-posterior region used the same sub-regional boundaries.

## Fluorescent in situ hybridization

Two-color fluorescent ISH was performed according to previous protocols (*Cembrowski et al., 2016*), with extensions for multiplexed approaches (described below). All probes were purchased from Advanced Cell Diagnostics (Hayward, CA) and were as follows: *Cplx1* (482531-C3) and *Gad1* (400951-C2) for two-color FISH, and *Nr4a2* (423351-T1), *Otof* (485671-T2), *Cdh13* (443251-T3), *Rorb* (444271-T4), *Adamts2* (806371-T5), *Prss23* (447921-T6), *Bdnf* (424821-T7), *Slc5a5* (487721-T8), *Slc17a7* (416631-T9), *Nnat* (432631-T10), *Negr1* (806361-T11), and *Cplx1* (482531-T12) for 12-channel multiplexed FISH.

For multiplexed FISH, 12 probes with unique detection tails were hybridized to tissue and subsequently amplified. Cleavable fluorophores specifically targeting probes 1–4 were then added, the tissue stained for DAPI, and coverslipped with ProLong Gold antifade mounting medium. Within 1–2 days, tissue was imaged with a Leica SP8 confocal microscope with a 63x objective. After this first round of imaging, the coverslip was removed by soaking slides in 20x SSC, and the fluorophores cleaved via two consecutive incubations in a 10% TCEP cleaving solution. Cleavable fluorophores specific to probes 5–8 were then added, and the tissue was coverslipped and imaged. This same procedure was then repeated for probes 9–12. Each round of imaging visualized DAPI in the 405 channel, and the subsequent four probes in either Alexa 488, ATTO 550, ATTO 647N, or Alexa 750.

After the three rounds of imaging to visualize probes 1–4 (round 1), 5–8 (round 2), and 9–12 (round 3), respectively, the images were computationally analyzed via registration, segmentation and quantification from in-house code in Fiji (*Schindelin et al., 2012*) (RRID:SCR_002285). The registration code aligned the three rounds of imaging in X, Y, and Z coordinates by taking the maximum projections of the DAPI signal from each round, and via a fast Fourier transform (FFT), identified offsets and translated the DAPI and probe signals from the second round in X and Y. To align in the Z-axis, a slice through the original DAPI stacks was taken and coordinates for translation in Z were once again identified via FFT offset. These steps were repeated to align the third round to the first. Segmentation involved taking the maximum projection of the DAPI signal for the first round and manually thresholding to select nuclei from background. Fiji's *Watershed* algorithm was run to separate any overlapping cells and then ROIs were selected using the *Analyze Particles* Fiji function. ROIs with an area less than 40 µm (typically reflecting partial cells) or greater than 200 µm (typically reflecting multiple abutting cells) were discarded, and the remaining ROIs dilated by a value of 5 µm to encompass the entire soma. Finally, quantification involved thresholding each of the 12 probe images to separate signal from background. Signals in the 750 channel (*Rorb, Slc5a5, Cplx1*) were quantified after applying a smoothing filter to reduce background. Using the ROIs selected from segmentation, the number of expressing pixels for each probe were summed and normalized by the total pixel area of the cell, producing values of counts per area (CPA).

The final dataset constituted three sections (anterior, intermediate, and posterior) from one WT mouse, with analysis of segmented, quantified cells proceeding as follows. First, non-excitatory neurons were removed by requiring *Slc17a7* expression (threshold = 0.004 CPA). Cell-to-cell correlations were calculated as Pearson coefficients. For binary *Cplx1* vs. *Negr1* phenotyping, based upon the strongly bimodal structure of *Cplx1* vs. *Negr1* expression (*Figure 6E*), a winner-take-all strategy was used to assign each excitatory neuron to either *Cplx1* or *Negr1* phenotypes by identifying the gene with greater expression. Excitatory neurons that expressed neither gene were excluded from any phenotype-associated analysis (n = 364 cells). For computing cell area, maximum projection images of *Slc17a7* from anterior, intermediate, and posterior sections were segmented in Fiji, similarly to DAPI. However, prior to thresholding a Gaussian blur filter was applied, but no watershed algorithm was used to prevent incorrect segmentation of cells due to the punctate in situ signal. ROIs with an area less than 75 µm$^2$ or greater than 450 µm$^2$ were discarded, and both *Negr1* and *Cplx1* signals then underwent quantification via the same approach mentioned previously. Clustering analysis pooled data from the anterior, intermediate, and posterior BLA sections. This aggregate data set was normalized such that CPA values for each cell added to equal 1. Hierarchical clustering of normalized data was performed in R, with a Euclidean distance metric and Ward's D2 clustering method. t-SNE (perplexity = 35) was used to visualize the data in dimensionally reduced space, with resultant plots colored according to section position or cluster identity.

## Viral tracing experiments and analysis

Mice underwent stereotaxic surgery, wherein either AAV-SL1-CAG-GFP or AAV-SL1-CAG-tdT was injected into a given region with known afferent input from the BLA; namely, the prefrontal cortex (A/P 1.7, M/L 0.60, D/V −2.3), nucleus accumbens (1.34, 1.0,−5.2), ventral hippocampus (−3.55, 3.25,−4.00), auditory cortex (−2.5, 4.5,−2.0), medial entorhinal cortex (−4.6, 3.3,−2.6) or retrosplenial cortex (−1.5, 0.5,−0.5). Dorsal/ventral coordinates were measured from the skull at site of injection (PFC, NAC, VHC), or from the pial surface (MEC, RSC). At each site, 100 nL of virus was injected over 2 min, and the needle remained at the injection site for 3 min post-injection. During surgery, mice were placed under isoflurane anesthesia, and received a local injection of bupivicane along the incision site. The day of surgery, and 2 days following surgery, mice received daily injections of

Metacam for post-operative care. Mice were perfused (10 mL of PBS, followed by 50 mL of 4% PFA) and brains extracted 5–9 days following surgery to permit sufficient expression of GFP or tdT.

Brains were cryoprotected in 30% sucrose in PBS for at least 48 hr, and sectioned at either 20 μm or 100 μm using a cryostat. Sections were mounted onto glass slides and either stored at −80℃ until use (20 μm sections), or were processed the following day (100 μm sections). Sections were counter-stained with DAPI (1:1000) for 10 min, and coverslipped with PVA-Dabco mounting medium. The location and size of viral-labeled cell bodies was determined manually. As with chromogenic in situ hybridization, locations of labeled cells were translated onto a common BLA template. For the PFC, MEC, ACX and VHC, the location of 28–168 cells were measured per animal. For the NAC, the location of 260–469 cells were measured per animal, given a higher density of virally labelled cells compared to the other injection sites. For the RSC, 12–32 cells were measured per animal, given a lower number of virally labeled cells compared to other injection sites.

To establish a null model to compute statistically enriched projections from LA or BA, we used *Slc17a7* (excitatory) neurons from LA and BA to establish chance estimates. For a given projection, N cells were selected at random, where N denotes the total number of cells from experimental data (summed across animals for a given projection). This random selection was repeated 1000 times, and the mean and 95% confidence intervals were calculated across iterations. Projections were denoted as statistically enriched if the empirically observed number fell outside of the 95% confidence interval.

## Fluorescence imaging

All mFISH histological images were acquired with a 63x objective on a SP8 white light laser confocal microscope (Leica Microsystems, Concord, Ontario, Canada), with z-stacks with step size of 0.35 μm were acquired for each round of imaging. Due to the large number of channels (12), final images are shown by overlaying each channel in order from highest to lowest expression rather than by merging channels, with all images shown depicting maximum intensity projections of the original stacks. Single-channel aligned mFISH images are available at https://figshare.com/projects/BLA_heterogeneity/87476. All other histological images were acquired using a 20x objective on an LSM 880 confocal microscope (Carl Zeiss Microscopy, Jena, Germany). Single optical sections or maximum intensity projections are shown, with the relevant regions tiled in X and Y dimensions as needed. In some cases, channels were postprocessed in Fiji (*Schindelin et al., 2012*), with brightness adjustments applied to the entire image and/or pseudocoloring.

## Statistical conventions

Central tendency and error bars denote mean ± SEM respectively, unless otherwise stated. Associated statistical parameters are reported within text or figure legends. Correlations were measured via Pearson correlation coefficients, and statistical significance was determined via unpaired Mann-Whitney U tests, unless otherwise stated. Statistical significance is denoted as follows: ns: $p \geq 0.05$; *: $p < 0.05$, **: $p < 0.01$, ***: $p < 0.001$.

## Acknowledgements

The authors thank members of the Cembrowski laboratory for helpful discussion and comments on the manuscript. Funding was provided by the University of British Columbia (Department of Cellular and Physiological Sciences, Djavad Mowafaghian Centre for Brain Health, and the Faculty of Medicine Research Office to M.S.C.), the Natural Sciences and Engineering Resource Council of Canada (RGPIN-2019–04507 to M.S.C.), the Canadian Institutes of Health Research (PJT-419798 to M.S.C.), the Canadian Foundation for Innovation (John R Evans Leaders Fund 38369 to M.S.C.), the Michael Smith Foundation for Health Research (SCH-2020–0383 to M.S.C.), and the Howard Hughes Medical Institute.

## Additional information

### Funding

| Funder | Grant reference number | Author |
| --- | --- | --- |
| Natural Sciences and Engineering Research Council of Canada | RGPIN-2019-04507 | Mark S Cembrowski |
| Canadian Institutes of Health Research | PJT-419798 | Mark S Cembrowski |
| Canada Foundation for Innovation | John R. Evans Leaders Fund 38369 | Mark S Cembrowski |
| University of British Columbia | | Mark S Cembrowski |
| Howard Hughes Medical Institute | | Mark S Cembrowski |
| Michael Smith Foundation for Health Research | SCH-2020-0383 | Mark S Cembrowski |

The funders had no role in study design, data collection and interpretation, or the decision to submit the work for publication.

### Author contributions

Timothy P O'Leary, Formal analysis, Investigation, Visualization, Methodology, Writing - review and editing; Kaitlin E Sullivan, Formal analysis, Visualization, Methodology, Writing - review and editing; Lihua Wang, Andrew L Lemire, Resources, Methodology, Writing - review and editing; Jody Clements, Resources, Data curation, Software, Writing - review and editing; Mark S Cembrowski, Conceptualization, Resources, Data curation, Formal analysis, Supervision, Funding acquisition, Investigation, Visualization, Methodology, Writing - original draft, Project administration, Writing - review and editing

### Author ORCIDs

Kaitlin E Sullivan (iD) https://orcid.org/0000-0001-8043-7111
Mark S Cembrowski (iD) https://orcid.org/0000-0001-8275-7362

### Ethics

Animal experimentation: Experimental procedures were approved by the Animal Care Committee at the University of British Columbia (A18-0267; A18-0285) and the Institutional Animal Care and Use Committee at the Janelia Research Campus (11-78).

### Decision letter and Author response

Decision letter https://doi.org/10.7554/eLife.59003.sa1
Author response https://doi.org/10.7554/eLife.59003.sa2

## Additional files

### Supplementary files

• Transparent reporting form

### Data availability

Raw and processed scRNA-seq datasets have been deposited in the National Center for Biotechnology Information (NCBI) Gene Expression Omnibus under GEO: GSE148866. Data underlying the results of this manuscript have been provided on FigShare (https://doi.org/10.6084/m9.figshare.c.5108165).

The following datasets were generated:

| Author(s) | Year | Dataset title | Dataset URL | Database and Identifier |
|---|---|---|---|---|
| Cembrowski M | 2020 | Discrete and continuous heterogeneity in the basolateral amygdala | https://www.ncbi.nlm. nih.gov/geo/query/acc. cgi?acc=GSE148866 | NCBI Gene Expression Omnibus, GSE148866 |
| Sullivan K,  Cembrowski M | 2020 | BLA heterpgeneity | https://doi.org/10.6084/ m9.figshare.c.5108165 | figshare, 10.6084/m9. figshare.c.5108165 |

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
