## [Decision Letter]

Thank you for submitting your article "Extensive and spatially variable within-cell-type heterogeneity across the basolateral amygdala" for consideration by *eLife*. Your article has been reviewed by three peer reviewers, one of whom is a member of our Board of Reviewing Editors, and the evaluation has been overseen by Laura Colgin as the Senior Editor. The following individual involved in review of your submission has agreed to reveal their identity: Benjamin W Okaty (Reviewer #2).

The reviewers have discussed the reviews with one another and the Reviewing Editor has drafted this decision to help you prepare a revised submission.

Summary:

The BLA is known to participate in a wide range of behaviors and neurons of the BLA are already known through tracing to connect to different regions. Here the authors ask whether there is heterogeneity in gene expression within different neurons of the BLA and conclude that these differences in gene expression may define neurons that participate in different BLA functions. Using scRNA-seq with in situ validation, the authors find transcriptomic separation of lateral from basal amygdala neurons as well as identifying less distinctly spatially segregated subpopulations of neurons in each of the two subdivisions. Overall this dataset will be helpful to researchers studying amygdala function.

Essential revisions:

The reviewers concurred on three points that they felt should be strengthened through text revision or data analysis to fully support the conclusions of the manuscript. I have included the full text of the reviewers' comments on these points including suggestions for further data or analyses because they offer insight into the nature of the concerns. These are just suggestions, however, and we invite the authors to decide how to respond.

1) The authors need to further justify the clustering parameters used.

One review said, "While I recognize it is not a trivial problem to identify clustering parameters to fully capture cell type diversity, I am concerned that the number of cells is in the dataset may be too low, which is limiting cluster resolution. Alternatively, the clustering parameters used for Figure 2 and beyond may not be optimized to capture more granular transcriptionally distinct cell types. Methods such as IKAP (https://www.biorxiv.org/content/10.1101/596817v1) have been developed to identify clustering parameters based on the particular dataset. Did the authors use this or some other method to arrive at their clustering parameters? Can they scientifically justify the parameters they chose?"

A second suggested, "Did the authors make any attempt to cluster cells based on mFISH CPA values of all 12 genes? If so, what did that look like; i.e. how well does this particular set of 12 genes serve to classify all six clusters identified by scRNAseq? Might it be possible to use this mFISH data to further characterize anterior/posterior or dorsal/ventral bias of finer-scale subclusters, rather than based solely on single marker genes? While the correlation analysis does a nice job of showing graded spatial heterogeneity, it would still be informative to see some sort of quantification of spatial bias of all the scRNA-seq subclusters, as inferred by mFISH."

2) A central and critical theme of the manuscript is the idea of continuous gradients of gene expression. However this concept raised two different sets of questions – one about the definition of cell types and one about the concept of continuous distributions in space.

a) While transcriptomic cell-typing is a powerful tool for neuron classification, cells that exist along a continuum in gene expression space may nonetheless constitute functionally distinct subtypes in ways that are not fully "reducible" to well-separated transcriptomic boundaries alone. Moreover, continuous expression gradients may become discretized at higher levels of regulation (e.g. a critical mRNA expression threshold may be required for protein expression, or gene/protein expression level may have some nonlinear relationship with higher-order cell phenotypes etc.). Could the authors perhaps give a more nuanced discussion of these possibilities? Specifically, I think it's important to explicitly make the distinction between transcriptomically defined neuron subtypes and subtypes defined by other criteria.

Certainly a desirable goal is to have a comprehensive measure of cell identity that subsumes multiple levels of description, and in ideal circumstances cellular transcriptome and other phenotypes may overlap in a clear-cut way. However, that may not always be the case. For example – when describing the results of Kim et al., 2016 with respect to there being functionally distinct subtypes of BLA neurons marked by expression of *Ppp1r1b* or *Rspo2*, the authors state: "In the context of our results here, we interpret these previous results as analyzing opposite extremes of a continuum; indeed, in our scRNA-seq data *Ppp1r1b* and *Rspo2* are enriched in opposite ends of the BA transcriptomic spectrum and do not conform to distinct subtypes. Thus, results predicated on analyzing these two populations may reflect relatively arbitrary divisions of a continuum, rather than examination of well-separated subtypes."

*Ppp1r1b* and *Rspo2* appear to be expressed at the highest levels in clusters BA1 and BA4, respectively. While the boundary between BA1 and BA4 in tSNE-space is more graded than discrete, these clusters are nonetheless separable by Louvain clustering, and random forest classification of subsampled data appears to fail mostly at the interface of these clusters, not at the "opposite extremes". Thus, if these two genes are enriched in "opposite ends of the BA transcriptomic spectrum", is it truly "arbitrary" to use these genes to parse BLA neurons into different subtypes? Moreover, Kim et al., 2016 found that neurons marked by one or the other gene exhibited different spatial biases, responded to different stimuli, and regulated different behaviors, all of which suggests that these indeed represent functionally distinct neuron subtypes.

b) My only concern relates to the way the authors discuss the concept of spatial gradients. I find the wording confusing both with respect to gene expression and with respect to the projections. One example is where the authors refer to a "continuous spectrum" of cell types defined by gene expression across the LA. This emerges again where it says "both the BA and LA exhibit variable cell-type identity that transformed continuously in space". However, especially since the authors are looking at one gene at a time, it is not clear it is quite continuous – which along with the word "graded" sounds to me like it goes from high to low so quantification would be required. With respect to the gene expression data that are analyzed by the multi-color in situ (which is a powerful experiment) it was unclear when the authors refer to "graded" versus "discrete" do they take into account levels of expression or is this just a +/- call above threshold? This is certainly a quantitative method but it is not clear in Figure 6 that the quantitative data were used. This would be nice to see and would likely make this concept more clear.

3) Two of the reviewers raised questions about BLA interneurons. The reviewers agreed that exploring interneuron transcriptomes was not essential in this paper but suggested that the authors clarify the following points:

a) The authors say they use a manual approach "that facilitates capture of excitatory neurons" but the methods offer no description of how that selectivity would be achieved other than via a careful dissection of a brain region primarily composed of those neurons. This should be clarified.

b) The paper largely ignores interneuron populations in and around the BLA. Including pericapsular GABAergic clusters. This is despite these cells being an integral component of BLA circuitry. To make the paper more comprehensive, the authors could include data from the 51 interneurons they collected, and provide some description of the cell types. If the authors cannot provide these data, it should be made explicitly clear in the text that these data are not included.

c) Subsection “Discrete separation in gene expression maps onto the lateral vs. basal amygdala nuclei” and Figure 3—figure supplement 1: If the authors want to make the strong statement that the "spill-over" is interneurons, they need to show that with double labeled in situ, not just argue that the pattern of expression is similar in number and general spatial distribution.

---

## [Author Response]

Essential revisions:The reviewers concurred on three points that they felt should be strengthened through text revision or data analysis to fully support the conclusions of the manuscript. I have included the full text of the reviewers' comments on these points including suggestions for further data or analyses because they offer insight into the nature of the concerns. These are just suggestions, however, and we invite the authors to decide how to respond.1) The authors need to further justify the clustering parameters used.One review said, "While I recognize it is not a trivial problem to identify clustering parameters to fully capture cell type diversity, I am concerned that the number of cells is in the dataset may be too low, which is limiting cluster resolution.

Generally, our plate-based manual capture approach is designed to produce high-read-depth transcriptomes, and thus is well-powered to detect gene-expression differences within profiled cells. Complementary approaches that are designed to survey a larger number of cells (e.g., droplet-based 10X Genomics Chromium) are more efficient at capturing rare cell types, but due to dropout effects and low depth, perform relatively poorly at detecting heterogeneity and subclusters within common cell types (see, e.g., “Direct Comparative Analysis of 10X Genomics Chromium and Smart-seq2”, Wang et al., bioRxiv, 2019). Thus, in principle, our approach is relatively well-positioned to reveal heterogeneity within classical types of the BLA.

In our revised manuscript, we now show this advantage is born out in practice. First, in examining transcriptomes of interneurons, we are able to identify and validate distinct interneuron subtypes despite only having tens of cells in each cluster (see new Figure 6—figure supplement 3; see also our response to comment 3). Requiring only tens of cells to resolve and validate separable clusters is also consistent with our previously published work (e.g., Cembrowski et al., 2018a; Cembrowski et al., 2018b). From these results, it seems unlikely that the >1,200 cells profiled from the BLA excitatory neuron population significantly limits cluster resolution.

To reinforce this, we also now perform a direct comparison to an existing, larger 10X

Genomics Chromium droplet-based dataset (see new Figure 2—figure supplement 2). Despite this droplet-based dataset having a great number of cells (n = 1,975 cells in this dataset; cf. 1,231 cells in our dataset), the relatively low-read-depth nature of this droplet-based dataset does not recover the clusters we discover in our work here.

Thus, multiple new lines of evidence suggest that our approach is well-suited to resolve transcriptomic heterogeneity of BLA excitatory neurons. Nonetheless, as with any transcriptomics study, there always exists the principle that acquiring more cells may illustrate further heterogeneity. We now mention this possibility, in conjunction with our above points, in our revised manuscript (see revised Discussion on “Graded spatial heterogeneity and functional implications”).

Alternatively, the clustering parameters used for Figure 2 and beyond may not be optimized to capture more granular transcriptionally distinct cell types. Methods such as IKAP (https://www.biorxiv.org/content/10.1101/596817v1) have been developed to identify clustering parameters based on the particular dataset. Did the authors use this or some other method to arrive at their clustering parameters? Can they scientifically justify the parameters they chose?"

Our approach for justifying our obtained scRNA-seq clusters is two-fold. First, we ensure agreement between clustering methods and other computational analyses. In our manuscript here, this took the form of ensuring clusters are in agreement with cluster-free visualizations of data in dimensionally-reduced space (e.g., Figure 2A, B; Figure 2—figure supplement 1), and that clusters are robust to down-sampling (e.g., Figure 2C). The parameters we chose in our manuscript here captured robust clusters across these complementary computational approaches.

Second, we next ensure that cell-type predictions from single-cell RNA-seq can be cross-validated by other experimental data. In our manuscript here, this took the form of cross-validation by the Allen Mouse Brain Atlas and mFISH (e.g., Figures 3, 5, 6). This approach, relying on different animals and methodologies (as well as laboratories, in the case of the Allen Mouse Brain Atlas), provides much stronger validation than can be provided by different analytical approaches applied to a fixed RNA-seq dataset.

In our revised manuscript, we now summarize this approach to cluster selection in our Methods section.

A second suggested, "Did the authors make any attempt to cluster cells based on mFISH CPA values of all 12 genes? If so, what did that look like; i.e. how well does this particular set of 12 genes serve to classify all six clusters identified by scRNAseq?”

We thank the reviewer for this suggestion, and in our revised manuscript, we now have performed clustering of mFISH data (new Figure 6—figure supplement 2). This analysis recapitulates the general organizational patterns we previously identified using scRNAseq and chromogenic ISH: two clusters recapitulate a BA vs. LA divide, and six clusters produces further graded heterogeneity within these discrete clusters. Thus, interestingly, this relatively small set of 12 genes is sufficient to reproduce the general organizational patterns of BLA heterogeneity.

Might it be possible to use this mFISH data to further characterize anterior/posterior or dorsal/ventral bias of finer-scale subclusters, rather than based solely on single marker genes? While the correlation analysis does a nice job of showing graded spatial heterogeneity, it would still be informative to see some sort of quantification of spatial bias of all the scRNA-seq subclusters, as inferred by mFISH.

In principle, we can use our mFISH data to infer fine-scale organization based on these 12 genes. However, we are wary to do this formally in the manuscript, as all of our analyses suggest within-BA and within-LA subclusters reflect elements of a continuum. Thus, emphasizing spatial domains of subclusters may lead the reader to incorrectly infer that subclusters reflect well-separated, distinct subtypes of cells. To avoid this potential confusion, we would prefer to avoid such a characterization.

2) A central and critical theme of the manuscript is the idea of continuous gradients of gene expression. However this concept raised two different sets of questions – one about the definition of cell types and one about the concept of continuous distributions in space.a) While transcriptomic cell-typing is a powerful tool for neuron classification, cells that exist along a continuum in gene expression space may nonetheless constitute functionally distinct subtypes in ways that are not fully "reducible" to well-separated transcriptomic boundaries alone. Moreover, continuous expression gradients may become discretized at higher levels of regulation (e.g. a critical mRNA expression threshold may be required for protein expression, or gene/protein expression level may have some nonlinear relationship with higher-order cell phenotypes etc.). Could the authors perhaps give a more nuanced discussion of these possibilities? Specifically, I think it's important to explicitly make the distinction between transcriptomically defined neuron subtypes and subtypes defined by other criteria.

The reviewer makes valuable points concerning the definition of cell types within the brain. In our revised manuscript, we have now included a Discussion section that explicitly makes the distinction between transcriptomically defined neuron subtypes and subtypes defined by other criteria (see “Cell-type classification: transcriptomics vs. other methodologies”). This section includes the specific points raised by the reviewer here.

Certainly a desirable goal is to have a comprehensive measure of cell identity that subsumes multiple levels of description, and in ideal circumstances cellular transcriptome and other phenotypes may overlap in a clear-cut way. However, that may not always be the case. For example, when describing the results of Kim et al., 2016 with respect to there being functionally distinct subtypes of BLA neurons marked by expression of Ppp1r1b or Rspo2, the authors state: "In the context of our results here, we interpret these previous results as analyzing opposite extremes of a continuum; indeed, in our scRNA-seq data Ppp1r1b and Rspo2 are enriched in opposite ends of the BA transcriptomic spectrum and do not conform to distinct subtypes. Thus, results predicated on analyzing these two populations may reflect relatively arbitrary divisions of a continuum, rather than examination of well-separated subtypes."Ppp1r1b and Rspo2 appear to be expressed at the highest levels in clusters BA1 and BA4, respectively. While the boundary between BA1 and BA4 in tSNE-space is more graded than discrete, these clusters are nonetheless separable by Louvain clustering, and random forest classification of subsampled data appears to fail mostly at the interface of these clusters, not at the "opposite extremes". Thus, if these two genes are enriched in "opposite ends of the BA transcriptomic spectrum", is it truly "arbitrary" to use these genes to parse BLA neurons into different subtypes? Moreover, Kim et al., 2016 found that neurons marked by one or the other gene exhibited different spatial biases, responded to different stimuli, and regulated different behaviors, all of which suggests that these indeed represent functionally distinct neuron subtypes.

The reviewer’s point is well-taken. Although ideally cell identity at the transcriptomic level would overlap with higher-order properties, there are many instances where this need not be the case. In our newly added section (“Cell-type classification: transcriptomics vs. other methodologies”; see also previous response point), we now discuss how discrete differences in higher-order phenotypes could emerge within the BA.

In particular, we suggest that this may occur for neuronal populations marked by the *Ppp1r1b* and *Rspo2* genes, wherein a graded transcriptomic identity could form the underpinning for more step-like changes in higher-order function and behavior.

b) My only concern relates to the way the authors discuss the concept of spatial gradients. I find the wording confusing both with respect to gene expression and with respect to the projections. One example is where the authors refer to a "continuous spectrum" of cell types defined by gene expression across the LA. This emerges again where it says "both the BA and LA exhibit variable cell-type identity that transformed continuously in space". However, especially since the authors are looking at one gene at a time, it is not clear it is quite continuous – which along with the word "graded" sounds to me like it goes from high to low so quantification would be required.

We thank the reviewer for noting the confusion surrounding the use of spatial gradients in the manuscript. We have now revised our wording to remove phrases like “continuous spectrum” and “graded” when referring to chromogenic ISH, which is a non-quantitative readout of gene expression. We also now explicitly note these caveats, and use them to motivate our mFISH analyses, which provide single-molecule resolution and quantitative data.

For projection cells, spatial gradients are shown in Figure 7, and further quantified in

Figure 7—figure supplement 1. To avoid confusion, we now explicitly note that these are “cellular” gradients, as projection cell quantification occurs at a cellular level and no molecular readout is present in these datasets.

With respect to the gene expression data that are analyzed by the multi-color in situ (which is a powerful experiment) it was unclear when the authors refer to "graded" versus "discrete" do they take into account levels of expression or is this just a +/- call above threshold? This is certainly a quantitative method but it is not clear in Figure 6 that the quantitative data were used. This would be nice to see and would likely make this concept more clear.

We apologize that our wording was not clearer, and confirm that quantitative data were indeed used in all mFISH analyses. In our revised Results section, we now explicitly note the quantitative nature of the data as we introduce this method.

3) Two of the reviewers raised questions about BLA interneurons. The reviewers agreed that exploring interneuron transcriptomes was not essential in this paper but suggested that the authors clarify the following points:

As described below, we now provide written clarifications on BLA interneurons. Moreover, although we concur that transcriptomic exploration of BLA interneurons was not necessarily essential in this work, we have nonetheless performed additional experiments and analysis examining BLA interneurons. This has resulted in two new supplementary figures in our revised manuscript (see new Figure 2—figure supplement 2 and Figure 6—figure supplement 3), which we believe further strengthen our manuscript. These results are described in depth in points (b) and (c) below.

a) The authors say they use a manual approach "that facilitates capture of excitatory neurons" but the methods offer no description of how that selectivity would be achieved other than via a careful dissection of a brain region primarily composed of those neurons. This should be clarified.

We now clarify this statement in our revised Results section, emphasizing that we primarily collect excitatory neurons due to their general prevalence, as well as their viability post-dissociation.

b) The paper largely ignores interneuron populations in and around the BLA. Including pericapsular GABAergic clusters. This is despite these cells being an integral component of BLA circuitry. To make the paper more comprehensive, the authors could include data from the 51 interneurons they collected, and provide some description of the cell types. If the authors cannot provide these data, it should be made explicitly clear in the text that these data are not included.

This point is well-taken and in our revised manuscript we now include analysis of the 51 interneurons we collected (see new Figure 6—figure supplement 3A-C). We have additionally complemented this new work with further analysis using chromogenic ISH (see new Figure 6—figure supplement 3D, E). Of particular note, this new analysis identifies two spatially distinct subtypes of interneurons, one of which is the pericapsular GABAergic clusters noted by this reviewer.

c) Subsection “Discrete separation in gene expression maps onto the lateral vs. basal amygdala nuclei” and Figure 3—figure supplement 1: If they authors want to make the strong statement that the "spill-over" is interneurons they need to show that with double labeled in situ, not just argue that the pattern of expression is similar in number and general spatial distribution.

As requested, in our revised manuscript we have now performed new experiments using double-label in situ hybridization. Results from these experiments are consistent with interneuron spill-over, as postulated in our original manuscript (see new Figure 3—figure supplement 2).